# Residual Pyramid Atrous Filtering Network with the Error Low-Rank Representation

## Abstract

Image filtering aims to eliminate perturbations and textures while preserving dominant structures, serving a pivotal role in various image processing tasks. More recently, significant advances in filtering techniques have been developed. However, existing approaches typically suffer from oversmoothing edges, gradient reversal, and halos. Such issues originate from the difficulty in striking an optimal trade-off between filtering multi-scale textures and preserving edges. Furthermore, deep learning-based filtering frameworks lack modules designed to capture features of different long-range dependence textures. Consequently, the task of filtering textures while maintaining edge integrity continues to pose a significant challenge. To address these issues, we propose a novel residual pyramid atrous filtering network (RPAFNet) that utilizes the error low-rank representation. Specifically, we introduce a lightweight dilated spatial convolution (LDSC) module for effectively extracting multi-scale texture features. To boost the reconstruction feature space, we propose a difference residual layer (DRL) module for connecting the encoder and decoder. Additionally, by employing low-rank approximation, we introduce a new non-convex optimization model, termed gradient error low-rank representation model (GELR), which effectively suppresses textures and preserves edges. This paper provides complete theoretical derivations for solving GELR and its convergence. Extensive experiments demonstrate that the proposed approach outperforms previous techniques in attaining an equilibrium between texture filtering and edge retention, as validated by both visual comparison and quantitative evaluation across various smoothing and downstream applications.

## 1 Introduction

Texture filtering is a core technique in computer graphics and vision, with applications ranging from detail enhancement (Zhong et al., 2023a) and compression artifact removal (Long et al., 2025) to tone mapping (Zhu et al., 2019). Its main objective is to suppress texture while preserving structural edges. However, the diversity and complexity of textures make this a persistent challenge. Existing methods fall into three main categories: local filtering (Gavaskar & Chaudhury, 2018), global optimization (He et al., 2023), and deep learning-based approaches (Shang et al., 2024). Local filters (Cho et al., 2014; Tomasi & Manduchi, 1998; Zhang et al., 2014a) use weighted pixel relationships to achieve image smoothing. However, since they have a fixed filter size, they often suffer from staircase artifacts, halo effects, and being unadaptive to multi-scale textures, like RGF (Zhang et al., 2014b) and MuGIF (Guo et al., 2018). Global optimization methods (Gudkov & Moiseev, 2020; He et al., 2023) typically convert the filtering task into a global optimization problem, which often has high computational costs. They struggle with multi-scale textures and also suffer from gradient reversal and halo artifacts. Deep learning approaches (Lu et al., 2018; Shang et al., 2024) leverage neural networks to learn feature representation from data to reconstruct smoothed images. These networks are limited to local information in filtering tasks. This hampers their ability to handle multi-scale textures. Figure 1 shows the case of handling multi-scale textures of existing approaches.

To address the challenge of handling multi-scale textures and preserving edges, we propose a novel residual pyramid atrous filtering network (RPAFNet) with the low-rank representation. Specifically, to extract multi-scale features, we introduce a lightweight dilated spatial convolution (LDSC) module to expand the receptive field, enabling the network to capture global and long-range texture information. To enhance the reconstruction feature space, we propose a difference residual layer (DRL)

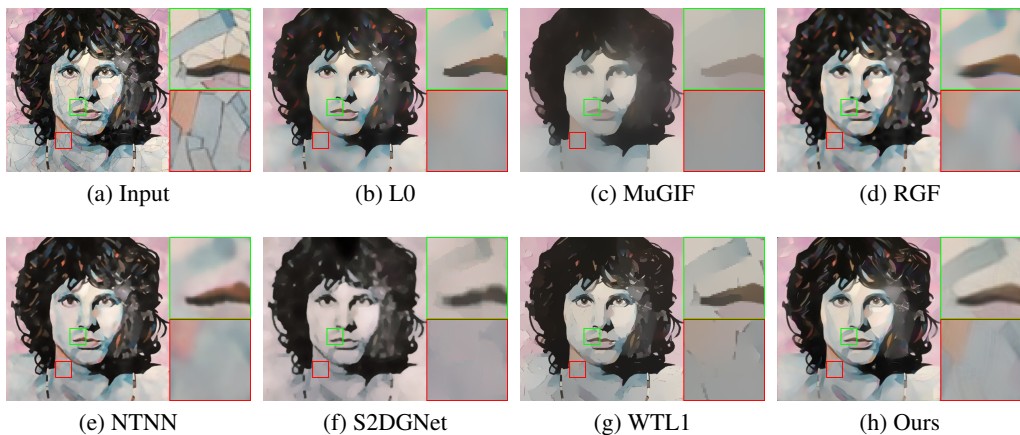

Figure 1: Comparison of handling multi-scale textures. (a) Input, smoothed results of (b) L0, (c) MuGIF, (d) RGF, (e) NTNN, (f) S2DGNet, (g) WTL1, and (h) Ours. It is hard to balance multi-scale texture removal and structural preservation for competing algorithms.

module in the proposed network. Finally, to overcome the over-smoothing edges and effectively suppress textures, we introduce a novel non-convex optimization filtering model with the error low-rank representation, which dynamically constrains the RPAFNet to preserve structural information. In a nutshell, the primary contributions of this work are concluded as follows: **(1)** We introduce the novel RPAFNet, including the proposed LDSC and DRL modules, which ensure effective handling of multi-scale and enhance the feature space for the reconstruction stage. **(2)** We propose a non-convex optimization model that utilizes a low-rank representation of the error map. The non-convex model dynamically constrains the RPAFNet to achieve texture removal and edge preservation. **(3)** Extensive experimental results demonstrate that our model outperforms state-of-the-art techniques in both visual quality and numerical performance across diverse smoothing applications.

Theoretical convergence derivations and additional downstream smoothing application experiments that have been omitted for space appear in the Appendix material.

## 2 RELATED WORK

**Local Filters.** Local filters smooth images by using nearby texture and structure information. Bilateral filtering (Tomasi & Manduchi, 1998), which weights neighboring pixels with Gaussian kernels, often causes gradient reversals and halo artifacts. Joint bilateral filtering (Cho et al., 2014) focus on low-structure regions to better extract textures. Edge-aware techniques (Xu & Wang, 2018) enhance structure preservation by incorporating edge weights. Recent advancements further refine window design and feature modeling: edge-aware windows reduce boundary interference (Xu & Wang, 2019), dynamic windows prevent texture-structure overlap (Pradhan & Patra, 2024), and histogram-based approaches improve texture-structure separation (Liu et al., 2020b). However, local filters rely on nearby pixel information, they struggle with multi-scale textures.

**Model Based Methods.** These algorithms formulate image smoothing as a global optimization problem, where data terms maintain similarity to the original image and regularization terms control texture suppression. Total variation (TV) (Rudin et al., 1992) minimizes image gradients to achieve smoothing but struggles with complex textures. Weighted least squares (WLS) (Farbman et al., 2008a) reduce artifacts more effectively but can introduce color shifts. Gradient minimization (Xu et al., 2011) controls non-zero gradients for improved smoothing. Relative total variation (RTV) (Xu et al., 2012) separates texture and structure via relative variation. Various prior-guided iterative methods have emerged. Locally adaptive models (Farbman et al., 2008a) and truncated Huber penalties (Li & Li, 2023) offer greater control over smoothing behavior. However, these methods still face challenges in balancing multi-scale texture smoothing and edge preservation, leading to the suffering from gradient reversal and halo artifacts.

**Learning Based Methods.** These approaches utilize neural networks for filtering that are typically categorized into two main types: supervised and unsupervised methods. Supervised models relied

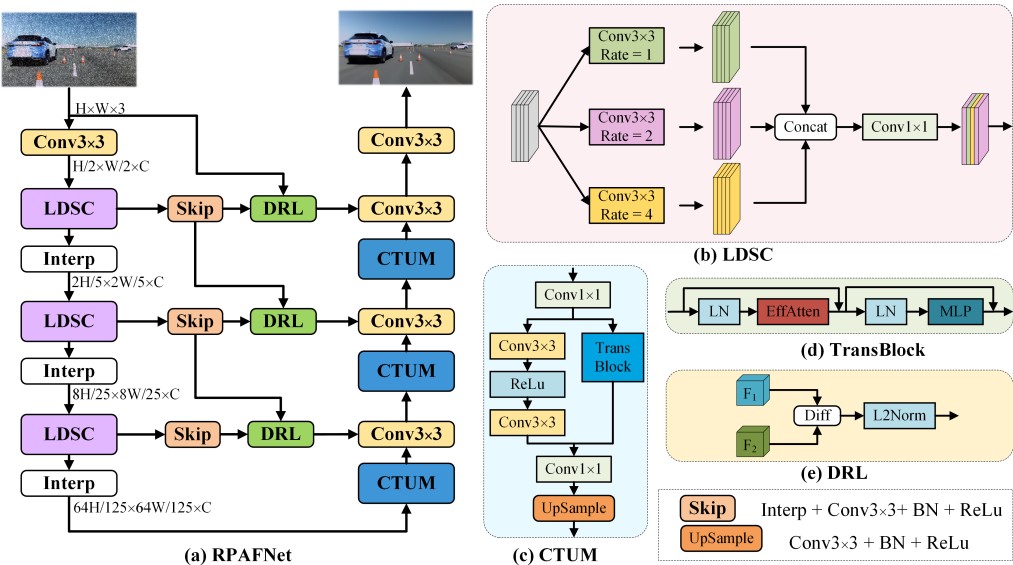

Figure 2: Main workflow of our proposed technique. (a) Main architecture of the Residual Pyramid Atrous Filtering Network (RPAFNet). (b) Structure of LDSC module, (c) Structure of CTUM module, (d) TransBlock module, (e) Structure of DRL module.

on ground truth data, such as attention-aware filters (Zhong et al., 2023b) and fully convolutional networks with large receptive fields (Chen et al., 2017b). E2H (Feng et al., 2021) improves results by jointly performing edge detection and structure-preserving smoothing using a tailored total variation loss. To reduce dependence on paired datasets, unsupervised methods emerged. Deep image prior (Ulyanov et al., 2018) uses randomly initialized networks as implicit priors. Later methods introduced input-dependent loss functions: bilateral texture loss in iterative networks (Jiang et al., 2024), weighted least squares in Deepwls (Yang et al., 2024c), and truncated norm-based regularization (Yang et al., 2024a). Despite progress, existing filtering networks lack modules for effectively extracting multi-scale texture features, limiting their effectiveness in capturing long-range dependencies.

## 3 METHODOLOGY

**Problem Description.** Given an input texture image $g$, and ground-truth $x$, we can consider the texture image to consist of image $x$ and texture layer image $T$, denoted as

$$g = x + T. \tag{1}$$

We aim to obtain the smoothed image $u$ via the proposed network with the input texture image $g$, denoted as:

$$u = f_\theta(g). \tag{2}$$

$f_\theta$ is the proposed residual pyramid atrous filtering network. However, image smoothing faces the big challenge in handling multi-scale textures. Motivated by the well-known dilated convolution (Chen et al., 2017a), we introduce a residual pyramid atrous filtering network to address this issue. To make our network capable of capturing long-range dependencies, we propose a lightweight dilated spatial convolution module to expand the receptive field in the encoder. To enrich the different levels of feature space for reconstruction, we introduce a difference residual layer module. The detailed architecture of the proposed residual pyramid atrous filtering network is shown in Figure 2. The following section introduces our designed network.

### 3.1 RESIDUAL PYRAMID ATROUS FILTERING NETWORK

To address the challenges posed by complex multi-scale textures, we propose a novel residual pyramid atrous filtering network, shown in Figure 2(a). RPAFNet utilizes an U-shaped architecture with a lightweight dilated spatial convolution module for encoding and a convolution transformer upsampling module for decoding. Skip connections between encoder and decoder layers are enhanced

using a difference residual layer to enrich the feature space. Downsampling is performed via the Interp module, which is a bilinear interpolation operator, progressively reducing feature size by a factor of 0.8. Upsampling in CTUM uses a $3 \times 3$ convolution layer, BatchNorm, and ReLU activation. LDSC module process features from the Interp module. Output features of LDSC undergo delta residual processing via the DRL module, which applies a subtraction operation and a L2Norm to refine details.

**Lightweight dilated spatial convolution module**. This model is designed for efficient feature extraction and enhanced texture awareness, as shown in Figure 2(b). The LDSC module is built based on the atrous convolution (Chen et al., 2017a), we leverage the dilated convolution (Yu & Koltun, 2016) as its unit convolution operator, as shown in Figure 2(b). This module consists of convolutional layers with varying dilation rates and scales, which is different from multi-scale dilated convolution in (Wang et al., 2019a). The LDSC module is simpler and lighter since it removes the BatchNorm and activation layers from these modules in (Chen et al., 2017a; Yu & Koltun, 2016; Wang et al., 2019a). The main differences lie in the use of $3 \times 3$ convolutional layers with dilation rates of 1, 2, and 4, followed by a concatenation operator and a $1 \times 1$ convolutional layer to unify the feature dimensions. Notably, the $3 \times 3$ convolutional layers with different dilation rates facilitate the extraction of features at different scales. This architecture allows the model to integrate multi-scale textural information.

**Convolution transformer upsampling module**. We propose the CTUM module to better handle image details. As shown in Figure 2(c), it processes features through two branches: a convolutional path and a Transformer path. The convolutional path uses two $3 \times 3$ convolutional layers with ReLU to extract local features. The Transformer path includes a TransBlock (Zhong et al., 2023b)(Figure 2(d)) made up of Layer Normalization (LN), Efficient Attention (EffAtten) (Shen et al., 2021), and a Multi-Layer Perceptron (MLP), along with skip connections to reduce overfitting. After both paths, features are refined using a $1 \times 1$ convolution and an upsampling module.

**Difference residual layer module**. This module extracts the features' differences from the previous layer and the skip layer, the architecture of which is shown in Figure 2(e). The difference map is passed to an L2Norm layer, which refers to the $L_2$-norm normalization of input features. The DRL module is designed to compensate for structural information during the decoding stage, thereby reducing the loss of edges and dominant structures.

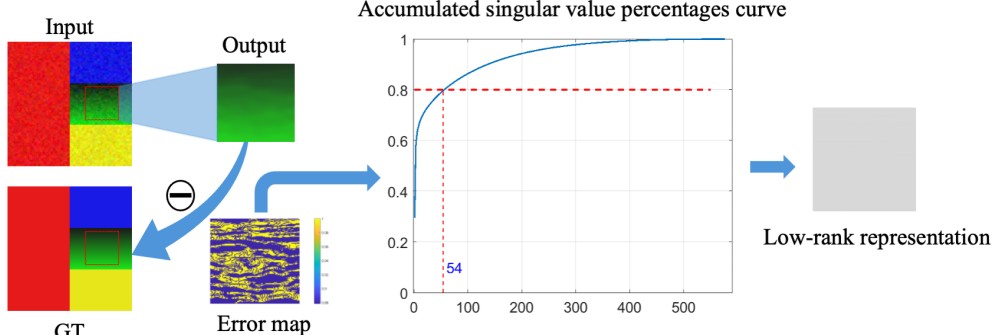

Figure 3: Pipeline of gradient error low-rank representation. The error map is the gradient error from the GT and output. And then employing a low-rank approximation on the error map suppresses textures.

### 3.2 GRADIENT ERROR LOW-RANK REPRESENTATION

To construct high-quality smoothed images, it needs to effectively filter textures and preserve edges. We aim to recover a smooth image $u$ from the textured image $g$. Ideally, $u$ is infinitely closer to $x$. Motivated by the merit of total variation $L_1$ regularization (Rudin et al., 1992; Li et al., 2025; Liu et al., 2024b) in edge preserving, we utilize $||\nabla u||_1$ regularization term to overcome the oversmoothing issue. To effectively suppress textures, we observe that low-rank approximation performs well in removing textures, which is inspired by low-rank approximation applied in denoising tasks, as shown in Figure 3. Therefore, we introduce a gradient error low-rank representation (GELR) model for integrated into the RPAFNet. It is worth noting that the gradient error is from between $u$ and $x$.

In a nutshell, the proposed gradient error low-rank representation model has two items, wirtten as

$$\min_u \quad \alpha||\nabla u||_1 + \beta||\nabla u - \nabla x||_r, \qquad \text{s.t.} \quad g = u + T. \tag{3}$$

$\alpha, \beta$ are positive penalty parameters. The second item is the low-rank approximation representation of the gradient error. $r$ is the selected rank in approximating the gradient error map. In this study, we can adjust values of $\alpha$ and $\beta$ to balance texture filtering and edge preservation.

To solve model equation 3, we introduce the auxiliary variables $\nabla u = d$ and $\nabla u - \nabla x = t$. The original problem equation 3 becomes

$$\min_{d,t} \quad \alpha||d||_1 + \beta||t||_r, \quad \text{s.t.} \quad g = u + T, \quad \nabla u = d, \quad \nabla u - \nabla x = t. \tag{4}$$

To address the proposed model effectively, we first introduce a key definition and a fundamental theorem.

**Definition 3.1 (Truncated Nuclear Norm** (Chen et al., 2024)**).** Given a matrix $Z \in \mathbb{R}^{m \times n}$, the truncated nuclear norm $||Z||_r$ is defined as:

$$||Z||_r = \sum_{i=r+1}^{\min\{m,n\}} \sigma_i(Z), \tag{5}$$

where $r = \lfloor \theta \min(m, n) \rfloor$, $\lfloor \cdot \rfloor$ denotes the largest integer that is less than or equal to input value. $\theta$ is the truncated rate, $\sigma_i$ denotes the singular values.

The truncated nuclear norm cannot be solved directly due to its non-convexity. Based on the analysis in (Xue et al., 2019), assuming that $Z$ has a singular value decomposition $Z = U\Sigma V^T$, where $U = (u_1, \cdots, u_m) \in \mathbb{R}^{m \times m}$, $\Sigma \in \mathbb{R}^{m \times n}$, and $V = (v_1, \cdots, v_n) \in \mathbb{R}^{n \times n}$. Therefore, the trunctated nuclear norm can become

$$||Z||_r = ||Z||_* - \max\left[\text{Tr}(AZB^t)\right], \quad \text{s.t.} \quad AA^T = I, \quad BB^T = I. \tag{6}$$

$A = (u_1, \cdots, u_r)^T \in \mathbb{R}^{r \times m}$ and $B = (v_1, \cdots, v_r)^T \in \mathbb{R}^{r \times n}$. $I \in \mathbb{R}^{r \times r}$ denotes the unit matrix. $\text{Tr}(\cdot)$ is the trace. We present the detailed derivation process in **appendix A.1**.

**Theorem 3.2.** *(Xue et al., 2019) For any given matrix $Q \in \mathbb{R}^{m \times n}$ with rank $r$. Then, the following problem has a unique closed-form solution, denoted as:*

$$Z_* = \arg\min_Z \mu||Z||_* + \frac{1}{2}||Z - Q||_2^2. \tag{7}$$

*It takes the form*

$$Z_* = SVT_{\mu,r}(Q) \in \mathbb{R}^{m \times n}, \tag{8}$$

*where $SVT_{\mu,r}(\cdot)$ is defined by*

$$SVT_{\mu,r}(Q) = U diag([\max(\sigma - \mu, 0)])V^T, \tag{9}$$

*where $U \in \mathbb{R}^{m \times r}$, $V \in \mathbb{R}^{r \times n}$, and $\sigma = (\sigma_1, \sigma_2, \sigma_3, \cdots, \sigma_r)^T \in \mathbb{R}^r$, which are obtained via the Singular Value Decomposition of Q. That means $Q = U diag(\sigma)V^T$. The detailed proof, see (Xue et al., 2019).*

### 3.3 OPTIMIZATION ALGORITHM

In this work, we use an alternating direction method of multipliers (ADMM) to solve the proposed model equation 4. The corresponding augmented Lagrange function is written as:

$$\mathcal{L}(u, d, t, \eta_d, \eta_t, T) = \frac{1}{2}||g - u - T||_2^2 + \alpha||d||_1 + \frac{\rho_1}{2}||\nabla u - d + \eta_d||_2^2$$
$$+ \beta||t||_r + \frac{\rho_2}{2}||\nabla u - \nabla x - t + \eta_t||_2^2, \tag{10}$$

where $\eta_d$ and $\eta_t$ are Lagrange multipliers, $\rho_1$, $\rho_2$ are Lagrange parameters. We split objective function equation 10 into the following subproblems, which means solving model equation 4 is

equivalent to solving equation 10 via iterative scheme. The all subproblems are listed as follows:

$$
\begin{cases}
u^{k+1} = \arg\min_u ||g - u - T^k||_2^2 + \frac{\rho_1}{2}||\nabla u - d^k + \eta_d^k||_2^2 + \frac{\rho_2}{2}||\nabla u - \nabla x - t^k + \eta_t^k||_2^2, \\
d^{k+1} = \arg\min_d \alpha||d||_1 + \frac{\rho_1}{2}||\nabla u^{k+1} - d + \eta_d^k||_2^2, \\
t^{k+1} = \arg\min_t \beta||t||_r + \frac{\rho_2}{2}||\nabla u^{k+1} - \nabla x - t + \eta_t^k||_2^2, \\
\eta_d^{k+1} = \eta_d^k + (\nabla u^{k+1} - d^{k+1}), \\
\eta_t^{k+1} = \eta_t^k + (\nabla u^{k+1} - \nabla x - t^{k+1}), \\
T^{k+1} = g - u^{k+1},
\end{cases}
\tag{11}
$$

where $k$ denotes the iteration number. Each subproblem is discussed in **appendix A.2**.

To ensure our designed network to learning texture and edge features, we utilize the proposed model to dynamically constrain RPAFNet training. Therefore, for solving the $u$-subproblem,

$$
u^{k+1} = \arg\min_u ||g - u - T^k||_2^2 + \frac{\rho_1}{2}||\nabla u - d^k + \eta_d^k||_2^2 + \frac{\rho_2}{2}||\nabla u - \nabla x - t^k + \eta_t^k||_2^2. \tag{12}
$$

We exploit the output of the proposed neural network to update $u$, meaning that $u^{k+1} = f_\theta(g)$. It is evident that equation 12 can be rewritten as a loss function, denoted as

$$
\mathcal{L}_1 = ||g - f_\theta(g) - T^k||_2^2 + \frac{\rho_1}{2}||\nabla f_\theta(g) - d^k + \eta_d^k||_2^2 + \frac{\rho_2}{2}||\nabla f_\theta(g) - \nabla x - t^k + \eta_t^k||_2^2, \tag{13}
$$

where $f_\theta$ represent the proposed neural network. $\mathcal{L}_1$ is one part of our total loss function, we also take the $\mathcal{L}_2$ loss, which is written as

$$
\mathcal{L}_2 = ||f_\theta(g) - x||_2^2 + \text{SSIM}(f_\theta(g), x), \quad \text{SSIM}(f_\theta(g), x) = 1 - \text{ssim}(f_\theta(g), x), \tag{14}
$$

where the ssim is the structural similarity index. The total loss function is

$$
\mathcal{L} = \lambda_1 \mathcal{L}_1 + \lambda_2 \mathcal{L}_2. \tag{15}
$$

$\lambda_1$ and $\lambda_2$ are two positive constants. The dynamic iterative strategy for constraining RPAFNet to training ensures the flexibility of our network's smoothing strength. The solution to each subproblem, computational complexity, and detailed global convergence proof of the non-convex optimization algorithm are presented in **appendix A.4**.

## 4 EXPERIMENTS

### 4.1 SETTINGS AND DATASETS

**Settings.** The proposed RPAFNet was driven to training via the proposed low-rank representation model, whose loss function is defined in equation 15 with $\lambda_1 = 0.7, \lambda_2 = 0.3$. Initially, parameters $\alpha, \beta$ are experimentally set to $0.4, 0.6$, respectively. While $\rho_1, \rho_2$ are assigned to 1 theoretically. Input images are resized into $512 \times 512$. We set the epoch number as 200, and batchsize is set to 4. The RPAFNet is updated via Adam optimizer with a learning rate 0.001. All experiments are conducted using PyTorch on a Ubuntu 20.04 server with two RTX 4090 GPUs. Our code will be available on github.

**Datasets.** We utilize the SPS (Feng et al., 2021) dataset to train RPAFNet, and compare performance on NKS(Xu et al., 2020) and ECS (Qi et al., 2024) datasets, which all have paired ground-truth smoothed images. Smoothing performance was assessed using PSNR and SSIM across the three datasets. We also utilized no-reference metrics: BRISQE (Mittal et al., 2012a), NIQE (Mittal et al., 2012b), PIQE (Venkatanath et al., 2015), ILNIQE (Zhang et al., 2015), and BLIINDS2 (Saad et al., 2012) to further evaluate performance for test images without paired ground-truth.

### 4.2 RESULTS ANALYSIS

We present a comparative analysis against state-of-the-art filtering techniques, including ILS (Liu et al., 2020a), L0 (Xu et al., 2011), L0L1 (Yang et al., 2022a), L1E (Yang et al., 2022b), PTF (Zhang et al., 2023), QWLS (Liu et al., 2024a), SEMF (Huang et al., 2023), WLS (Farbman et al., 2008b), CSGIS (Wang et al., 2022), E2H (Feng et al., 2021), Deepwls (Yang et al., 2024c), NTNN (Zhu et al., 2024), S2DGNet (Qi et al., 2024), and WTL1 (Yang et al., 2024b). For non-deep traditional methods, hyperparameters are configured according to the settings reported in their original papers

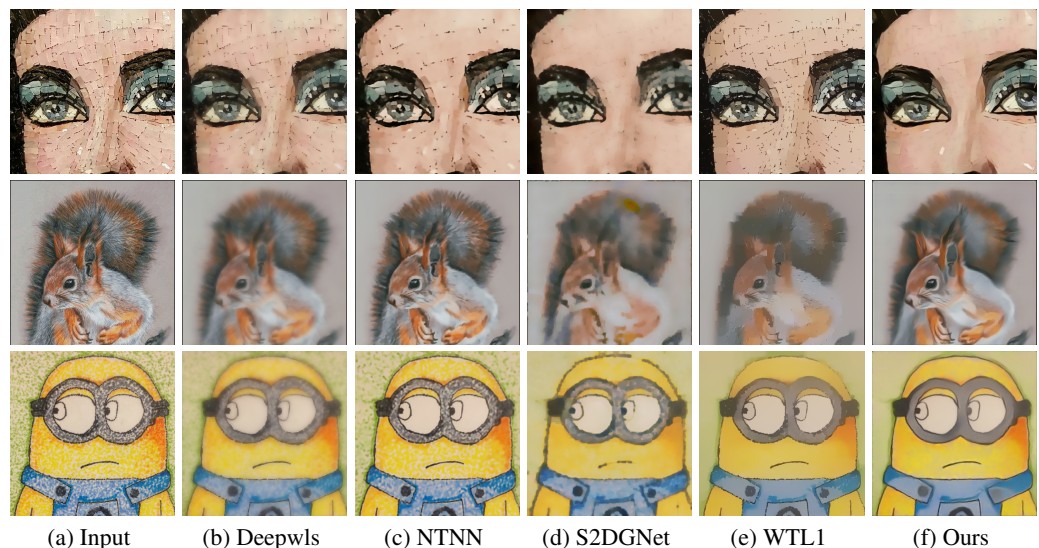

| (a) Input | (b) Deepwls | (c) NTNN | (d) S2DGNet | (e) WTL1 | (f) Ours |

Figure 4: Image filtering for competing approaches. Results of (a) input images, (b) Deepwls, (c) NTNN, (d) S2DGNet, (e) WTL1, (f) Ours, respectively. It is evident that the proposed model obtains the best visual effects.

and tuned to enhance performance. We utilize pre-trained models released by authors for deep learning-based approaches.

Figure 4 and Table 1 show results of different methods on three real-world images. RPAFNet outperforms current state-of-the-art methods in removing textures while preserving edges. In the first row of Figure 4, Deepwls, NTNN, and WTL1 fail to remove textures effectively. S2DGNet performs better, but our method produces the best visual results. In the second and third rows, Deepwls over-smooths the images, and NTNN fails in these two cases. S2DGNet and WTL1 also struggle to preserve edges. In contrast, our method achieves both effective smoothing and structure preservation. Table 1 reports four no-reference quality metrics, where RPAFNet consistently achieves the top scores.

Table 1: No-reference metric values on Figure 4.

| Methods | BRISQUE ↓ | NIQE ↓ | PIQE ↓ | ILNIQE ↓ | Mean ↓ |
|---|---|---|---|---|---|
| L0L1 (2022) | 60.899 | 7.916 | 75.196 | 58.167 | 50.545 |
| PTF (2023) | 40.235 | 6.128 | 72.312 | 27.167 | 36.461 |
| QWLS (2024) | 44.581 | 5.448 | 81.209 | 35.500 | 41.685 |
| CSGIS (2022) | 28.528 | 4.694 | 46.798 | 14.333 | 23.588 |
| E2H (2021) | 31.587 | 4.530 | 64.916 | 28.333 | 32.342 |
| Deepwls (2023) | 51.011 | 5.057 | 82.439 | 31.833 | 42.585 |
| NTNN (2024) | 43.225 | 5.021 | 80.325 | 42.000 | 42.643 |
| WTL1 (2024) | 50.419 | 7.583 | 81.208 | 52.667 | 47.969 |
| Ours | **19.725** | **4.491** | **46.768** | **12.333** | **20.829** |

To demonstrate RPAFNet strong edge-preservation ability, we present enlarged areas and their corresponding 1D smoothed signals in Figure 5. The blue line represents the input signal, and the red line shows the smoothed result. Key areas are highlighted with red arrows. CSGIS and E2H fail to remove textures cleanly. Deepwls, NTNN, and WTL1 overly filter edges, as seen in the peaks marked by the first arrows. S2DGNet performs reasonably well but introduces staircase artifacts. In contrast, our method effectively removes textures while preserving sharp and clean edges.

## 4.3 ABLATION STUDY

**LDSC module**. To evaluate the effectiveness of the LDSC module in handling multi-scale textures, we perform an ablation study, as shown in Figure 6. Without the LDSC module, the baseline network struggles to remove multi-scale textures as shwon in Figure 6(b). It still contains noticeable

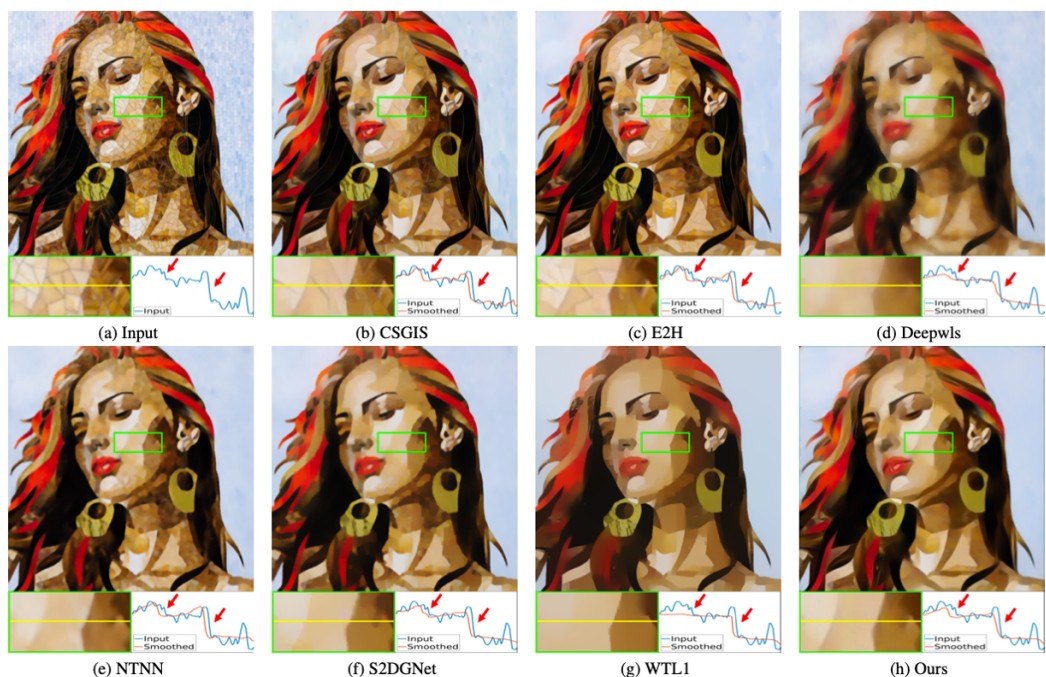

Figure 5: Texture removal comparisons. (a) The input image, results of (b) CSGIS, (c) E2H, (d) Deepwls, (e) NTNN, (f) S2DGNet, (g) WTL1, and (h) Ours.The right-bottom part of each image is the 1D signal smoothed result corresponding to the yellow line in the green marked box. The blue line is the input 1D signal, while the red is the smoothed result. We note that our method has a better-smoothed output than other techniques. Meanwhile, the proposed model keeps better edges.

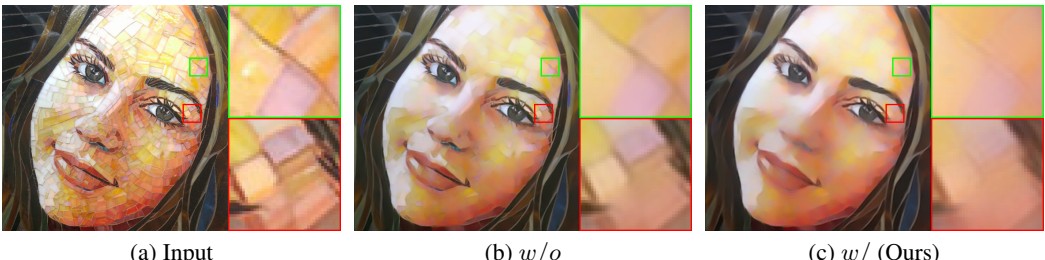

Figure 6: Ablation study on the LDSC module. (a) The input image, (b) $w/o$ denotes the baseline network without LDSC module, while (c) $w/$ denotes our full network. The significant effects of the LDSC module on textures can be seen in these enlarged areas.

Table 2: No-reference metrics on the LDSC module ablation study.

| Methods | Metrics | BRISQUE ↓ | PIQE ↓ | ILNIQE ↓ |
|---------|---------|-----------|--------|----------|
| Baseline | $w/o$ | 29.786 | 45.096 | 21.000 |
| Ours | $w/$ | **27.086** | **35.568** | **17.500** |

textures, referring to highlighted and enlarged regions. In contrast, RPAFNet successfully removes these textures, as illustrated in Figure 6(c). Meanwhile, Table 2 reports no-reference quality metrics corresponding to this ablation study. Both the visual results and metric values demonstrate the LDSC module effectiveness in smoothing multi-scale textures.

**DRL module**. We conduct an ablation study to validate the capability of the DRL module in enriching the feature space for reconstruction, as shown in Figure 7. The output from RPANet retains more fine details than that of the baseline network, achieving a PSNR of 27.43 and an SSIM of 0.9065. Both the visual results and quantitative metrics indicate that the DRL module enhances the feature space, allowing for the preservation of more content.

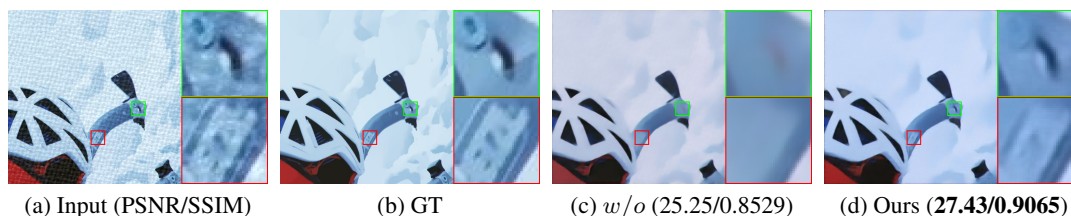

(a) Input (PSNR/SSIM)  (b) GT  (c) $w/o$ (25.25/0.8529)  (d) Ours (**27.43/0.9065**)

Figure 7: Ablation experiments for the DRL module. (a) Input (b) GT, (c) $w/o$ denotes smoothed result of removing DRL module in the proposed network, (d) smoothed image of RPANet. Full network obtains the best index values and visual effects.

Table 3: No-reference metrics on loss functions ablation study.

| Methods | Metrics | **BRISQUE ↓** | **PIQE ↓** | **BLIINDS2 ↑** |
|---|---|---|---|---|
| Baseline | $\lambda_1 = 0$ | 30.457 | 44.326 | 33.963 |
| RPAFNet | $\lambda_1 \neq 0$ | **21.172** | **37.514** | **52.748** |

**Parameters** $\lambda_1, \lambda_2$ **and** $\alpha, \beta$. We have confirmed the parameters' impact on smoothing performance by ablation studies. Experimental results of $\lambda_1 = 0$ have been reported in Table 3. The $\mathcal{L}_1$ loss is optimized iteratively using the ADMM algorithm, whereas $\mathcal{L}_2$ is directly optimized within an end-to-end framework. Thus, evaluating $\mathcal{L}_1$ also implicitly assesses the impact of the optimization algorithm, whose corresponding visual comparison and different values of $\lambda_1, \lambda_2$ have been shown in **appendix B**. To confirm the values of $\alpha, \beta$, we have conducted a series of experiments for each of them from 0.1 to 1.0. Quantitative numerical results are shown in Table 4.

Table 4: Quantitative results of different values for $\alpha, \beta$.

| $(\alpha, \beta)$ | (0.1,0.9) | (0.2,0.8) | (0.3,0.7) | (0.4,0.6) | (0.5,0.5) |
|---|---|---|---|---|---|
| **BRISQUE↓** | 49.761 | 45.746 | 35.075 | **27.448** | 29.539 |
| **NIQE ↓** | 7.593 | 6.219 | 5.827 | **4.692** | 5.146 |
| $(\alpha, \beta)$ | (0.6,0.4) | (0.7,0.3) | (0.8,0.2) | (0.9,0.1) | (1.0,0) |
| **BRISQUE↓** | 30.471 | 35.841 | 38.775 | 41.577 | 45.381 |
| **NIQE↓** | 4.922 | 5.792 | 6.933 | 7.891 | 10.273 |

**CTUM module.** CTUM module enables the fusion of both local and global representations, thereby enriching the feature space. We conduct an ablation study to confirm its reconstruction performance in the deconder stage, as shown in **appendix B**. The best index values demonstrate that the CTUM module effectively preserves image fines.

## 5 CONCLUSIONS AND LIMITATTIONS

This work introduces a novel smoothing network with integrates gradient error low-rank representation, named the residual pyramid atrous filtering network (RPAFNet). The LDSC module serves as a tool for effectively extracting multi-scale texture features. The proposed DRL module enhances the reconstruction feature space to enable RPAFNet to keep essensail fines. We introduce a novel non-convex gradient error low-rank representation model for dynamically constraining RPAFNet to learning discrimination between textures and edges. The solution of the proposed model is supported by a complete theoretical guarantee with the ADMM algorithm. Extensive experiments, including smoothing and downstream applications, demonstrate that RPAFNet outperforms state-of-the-art approaches in mitigating JPEG compression blocks, gradient reversal, and halos. Whether deep learning or non-deep learning filtering techniques, RPAFNet consistently achieves a superior balance between filtering multi-scale textures and edge preservation.

**Limitations**. Supervised deep learning-based filtering techniques, including our RPAFNet, have a common limitation: their performance upper is limited by training pairs. A promising direction for future work would be to design a self-supervised framework for filtering multi-scale textures. Meanwhile, although our RPAFNet achieves superior performance in handling multi-scale textures, it has constraints when dealing with low contrast textures, which means texture color close to that of the background. Exploring the potential of different color space types' impact could provide valuable insights into achieving more effective texture filtering while maintaining edges.

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

## THE USE OF LARGE LANGUAGE MODELS

We declare that we just used the large language models to improve sentences and polish in this manuscript.

## SUMMARY

This technical appendix offers a comprehensive theoretical analysis of our model, including a detailed examination of its convergence properties. Additionally, it presents both visual and numerical results for smoothing downstream tasks. The structure of this appendix is organized as follows. Section A presents a mathematical analysis of the proposed non-convex problem, which includes the derivation process of the truncated nuclear norm and numerical solution to the gradient error prior model. Meanwhile, we also analyze the convergence of the proposed optimization algorithm. Section B shows additional ablation study experimental results. Section C provides analysis of three additional application experimental results, including details manipulation, image stylization, and artifacts filtering. We also present addittional experimental results in Section C.

## A  THEORETICAL ANALYSIS OF GELR MODEL

This section mainly presents the mathematical analysis on the truncated nuclear norm, solution to the non-convex optimization problem, and convergence analysis of our optimization algorithm.

### A.1  DERIVATION OF TRUNCATED NUCLEAR NORM

First of all, we review the proposed non-convex optimization problem, which is denoted as:

$$\min_{d,t} \quad \alpha||d||_1 + \beta||t||_r,$$
$$\text{s.t.} \quad g = u + T, \quad \nabla u = d, \quad \nabla u - \nabla x = t. \tag{16}$$

To better analyze the proposed non-convex problem, we separate the low-rank prior terms for detailed analysis, denoted as:

$$\min_{t} \quad \beta||t||_r,$$
$$\text{s.t.} \quad \nabla u - \nabla x = t. \tag{17}$$

Since $||t||_r$ is non-convex, it is not easy to solve directly, then we have the following theorem (Hu et al., 2012).

**Theorem A.1** ((Hu et al., 2012)). *For any given matrix $X \in \mathbb{R}^{m \times n}$, any matrices $A \in \mathbb{R}^{r \times m}$, $B \in \mathbb{R}^{r \times n}$. Such that $AA^T = I_{r \times r}$, $BB^T = I_{r \times r}$. For any nonnegative integer $r$ ($r \leq \min(m,n)$), we have*

$$Tr(AXB^T) \leq \sum_{i=1}^{r} \sigma_i(X).$$

Therefore, for the proposed model, let $X = t$, and then the detailed proof is as follows.

**Proof.** By the Von Neumann's trace inequality, we can have

$$\text{Tr}(AtB^T) = \text{Tr}(tB^T A) \leq \sum_{i=1}^{\min(m,n)} \sigma_i(t)\sigma_i(B^T A), \tag{18}$$

where $\sigma_1(t) \geq \cdots \geq \sigma_{\min(m,n)}(t) \geq 0$. Since $rank(A) = r$ and $rank(B) = r$, then $rank(B^T A) = s \leq r$. For $i \leq s$, we can get $\sigma_i(B^T A) \geq 0$. $\sigma_i^2(B^T A)$ is the $i$-th eigenvalue of $B^T AA^T B = B^T B$, which is also the eigenvalue of $BB^T = I_{r \times r}$. Therefore,

$\sigma_i(B^T A) = 1, \forall i \geq s$, and others are 0. Thereby, we can get the follows:

$$
\begin{aligned}
\sum_{i=1}^{\min(m,n)} & \sigma_i(t)\sigma_i(B^T A) \\
&= \sum_{i=1}^{s} \sigma_i(t)\sigma_i(B^T A) + \sum_{i=s+1}^{\min(m,n)} \sigma_i(t)\sigma_i(B^T A) \\
&= \sum_{i=1}^{s} \sigma_i(t) \cdot 1 + \sum_{i=s+1}^{\min(m,n)} \sigma_i(t) \cdot 0 \\
&= \sum_{i=1}^{s} \sigma_i(t).
\end{aligned}
\tag{19}
$$

Since $s \leq r$ and $\sigma_i(t) \geq 0$, we have

$$
\sum_{i=1}^{s} \sigma_i(t) \leq \sum_{i=1}^{r} \sigma_i(t).
\tag{20}
$$

Combining equation 18 and equation 19, we can get

$$
\text{Tr}(AtB^T) \leq \sum_{i=1}^{s} \sigma_i(t) \leq \sum_{i=1}^{r} \sigma_i(t).
\tag{21}
$$

Assuming that $t$ has its singular value decomposition $t = U\Sigma V^T$, where $U = (u_1, \cdots, u_m) \in \mathbb{R}^{m \times m}$, $\Sigma \in \mathbb{R}^{m \times n}$, and $V = (v_1, \cdots, v_n) \in \mathbb{R}^{n \times n}$. And when $A = (u_1, \cdots, u_r)^T$ and $B = (v_1, \cdots, v_r)^T$, we have:

$$
\begin{aligned}
\text{Tr}(AtB^T) &= \text{Tr}((u_1, u_2, u_3, \cdots, u_r)^T t (v_1, v_2, v_3, \cdots, v_r)) \\
&= \text{Tr}((u_1, u_2, u_3, \cdots, u_r)^T U\Sigma V^T (v_1, v_2, v_3, \cdots, v_r)) \\
&= \text{Tr}(((u_1, u_2, u_3, \cdots, u_r)^T U)\Sigma(V^T (v_1, v_2, v_3, \cdots, v_r))) \\
&= \text{Tr}\left( \begin{pmatrix} I_r & 0 \\ 0 & 0 \end{pmatrix} \Sigma \begin{pmatrix} I_r & 0 \\ 0 & 0 \end{pmatrix} \right) \\
&= \text{Tr}(diag(\sigma_i(t), \cdots, \sigma_i(t), 0, \cdots, 0)) \\
&= \sum_{i=1}^{r} \sigma_i(t)
\end{aligned}
\tag{22}
$$

Combining equation 21 and equation 22, we can get

$$
\max_{AA^T=I, BB^T=I} \text{Tr}(AtB^T) = \sum_{i=1}^{r} \sigma_i(t).
\tag{23}
$$

Then

$$
\begin{aligned}
||t||_* &- \max_{AA^T=I, BB^T=I} \text{Tr}(AtB^T) \\
&= \sum_{i=1}^{\min(m,n)} \sigma_i(t) - \sum_{i=1}^{r} \sigma_i(t) = \sum_{i=r+1}^{\min(m,n)} \sigma_i(t) \\
&= ||t||_r.
\end{aligned}
\tag{24}
$$

$\blacksquare$

In summary, the non-convex optimization problem equation 17 can be rewritten as

$$
\arg\min_{t} ||t||_* - \max_{AA^T=I, BB^T=I} \text{Tr}(AtB^T)
$$
$$
\text{s.t.} \quad \nabla u - \nabla x = t.
\tag{25}
$$

The above problem is the same as Eq.(5) of the main manuscript. Then

$$\arg\min_t ||t||_* - \max_{AA^T=I,BB^T=I} \text{Tr}(AtB^T)$$
$$+ \frac{\rho}{2}||t - (\nabla u - \nabla x + \eta_t)||_2^2, \tag{26}$$

where $\rho$ is the balance positive constant, $\eta_t$ is the Lagrange multiplier. According to (Zhu et al., 2024), we can get the concise form of equation 26, denoted as

$$\arg\min_t ||t||_* + \frac{\rho}{2}||t - (\nabla u - \nabla x + \eta_t - A^T B)||_2^2, \tag{27}$$

Therefore, we can use **Theorem 3.2** to solve the non-convex optimization problem equation 27.

## A.2 NUMERICAL SOLUTION TO GELR MODEL

The proposed gradient error prior model can be expressed as the form in equation 16, we use the alternating direction method of multipliers (ADMM) to solve it. The corresponding augmented Lagrange function can be splited into subproblems, denoted as follows:

$$\begin{cases} u^{k+1} = \arg\min_u ||g - u - T^k||_2^2 + \frac{\rho_1}{2}||\nabla u - d^k + \eta_d^k||_2^2 + \frac{\rho_2}{2}||\nabla u - \nabla x - t^k + \eta_t^k||_2^2, \\ d^{k+1} = \arg\min_d \alpha||d||_1 + \frac{\rho_1}{2}||\nabla u^{k+1} - d + \eta_d^k||_2^2, \\ t^{k+1} = \arg\min_t \beta||t||_r + \frac{\rho_2}{2}||\nabla u^{k+1} - \nabla x - t + \eta_t^k||_2^2, \\ \eta_d^{k+1} = \eta_d^k + (\nabla u^{k+1} - d^{k+1}), \\ \eta_t^{k+1} = \eta_t^k + (\nabla u^{k+1} - \nabla x - t^{k+1}), \\ T^{k+1} = g - u^{k+1}, \end{cases} \tag{28}$$

where $k$ denotes the iteration number. Each subproblem are discussed as follows.

**Update** $u^{k+1}$ by

$$u^{k+1} = \arg\min_u ||g - u - T^k||_2^2 + \frac{\rho_1}{2}||\nabla u - d^k + \eta_d^k||_2^2$$
$$+ \frac{\rho_2}{2}||\nabla u - \nabla x - t^k + \eta_t^k||_2^2. \tag{29}$$

It is obvious that this subproblem has a closed-solution, and its first-order optimal condition is

$$T^k - g - \rho_1 \nabla^T (d^k - \eta_d^k) - \rho_2 \nabla^T (\nabla x + t^k - \eta_t^k)$$
$$+ (1 + \rho_1 \nabla^T \nabla + \rho_2 \nabla^T \nabla)u = 0. \tag{30}$$

Then, we have

$$g - T^k + \rho_1 \nabla^T (d^k - \eta_d^k) + \rho_2 \nabla^T (\nabla x + t^k - \eta_t^k)$$
$$= (1 + \rho_1 \nabla^T \nabla + \rho_2 \nabla^T \nabla)u. \tag{31}$$

According to the Fourier convolution theorem, we conduct Fourier transform on equation 31 and obtain

$$u^{k+1} = \mathcal{F}^{-1}\left( \frac{\mathcal{F}(g - T^k) + \rho_1 \mathcal{F}(\nabla^T (d^k - \eta_d^k))}{\mathcal{F}(1) + (\rho_1 + \rho_2)\mathcal{F}(\nabla^T \nabla)} \right) + \mathcal{F}^{-1}\left( \frac{\rho_2 \mathcal{F}(\nabla^T (\nabla x + t^k - \eta_t^k))}{\mathcal{F}(1) + (\rho_1 + \rho_2)\mathcal{F}(\nabla^T \nabla)} \right). \tag{32}$$

$\mathcal{F}$ and $\mathcal{F}^{-1}$ denote fast Fourier transform and inverse fast Fourier transform respectively.

**Update** $d^{k+1}$ by

$$d^{k+1} = \arg\min_d \alpha||d||_1 + \frac{\rho_1}{2}||\nabla u^{k+1} - d + \eta_d^k||_2^2, \tag{33}$$

The subproblem of those can be solved via the soft-thresholding skrinkage, denoted as

$$d^{k+1} = \textbf{shrink}(\nabla u^{k+1} + \eta_d^k, \frac{1}{\rho_1}). \tag{34}$$

---

**Algorithm 1** GELR-ADMM

---

**Input:** Image $g$, $x$, $\alpha$ $\beta$, $\rho_1$, $\rho_2$, $\theta$, $K$.
**Initialization:** $T^0, t^0, d^0, \eta_t^0, \eta_d^0 = \mathbf{0}$, $u^0 = g$.
  1: **for** $k = 1 : K$ **do**
  2:    Update $u^{k+1}$ via Eq. equation 32;
  3:    Update $d^{k+1}$ by Eq. equation 34;
  4:    Update $t^{k+1}$ via Eq. equation 40;
  5:    Update $\eta_d^{k+1}$ and $\eta_t^{k+1}$ via Eq. equation 41;
  6:    Update $T^{k+1}$ by Eq. equation 42;
  7: **end for**
**Output:** $u$.

---

**shrink** function is

$$\mathbf{shrink}(\nabla u^{k+1} + \eta_d^k, \frac{1}{\rho_1}) = $$
$$\mathrm{sign}(\nabla u^{k+1} + \eta_d^k) \cdot \max(|\nabla u^{k+1} + \eta_d^k| - \frac{1}{\rho_1}, 0), \tag{35}$$

where $\mathrm{sign}(\cdot)$ is the Signum function.

**Update $t^{k+1}$ by**

$$t^{k+1} = \arg\min_t \beta ||t||_r + \frac{\rho_2}{2}||\nabla u^{k+1} - \nabla x - t + \eta_t^k||_2^2, \tag{36}$$

according to the descriptions of **Definition 3.1** and **Theomery 3.2**, we can solve problem equation 36 via $\mathrm{SVT}_{\mu,r}$. Therefore, $t-$subproblem can be rewritten as

$$t^{k+1} = \arg\min_t \frac{\beta}{\rho_2}||t||_* + \frac{1}{2}||t - [\nabla u^{k+1} - \nabla x - t - \frac{\beta}{\rho_2}A^T B + \eta_t^k]||_2^2, \tag{37}$$

where $A$ and $B$ are obtained by the singular value decomposition of matrix $t$. Let $Q = \nabla u^{k+1} - \nabla x - t - \frac{\beta}{\rho_2}A^T B + \eta_t^k$, we have

$$t^{k+1} = \arg\min_t \frac{\beta}{\rho_2}||t||_* + \frac{1}{2}||t - Q||_2^2. \tag{38}$$

We have the unique closed-form solution is

$$t^{k+1} = \mathrm{SVT}_{\frac{\beta}{\rho_2},r}(Q). \tag{39}$$

Then, we have

$$t^{k+1} = \mathrm{SVT}_{\frac{\beta}{\rho_2},r}(Q) = U diag[max(\sigma - \frac{\beta}{\rho_2}, 0)]V^T, \tag{40}$$

where $U \in \mathbb{R}^{r \times m}$, $V \in \mathbb{R}^{r \times n}$, and $\sigma = (\sigma_1, \cdots, \sigma_r)^T \in \mathbb{R}^r$ are from the Singular Value Decomposition of $Q$.

**Update $\eta_d^{k+1}$ and $\eta_t^{k+1}$ by**

$$\begin{cases} \eta_d^{k+1} = \eta_d^k + (\nabla u^{k+1} - d^{k+1}), \\ \eta_t^{k+1} = \eta_t^k + (\nabla u^{k+1} - \nabla x - t^{k+1}), \end{cases} \tag{41}$$

the two Lagrange multipliers can be updated directly.

**Update $T^{k+1}$ by**

$$T^{k+1} = g - u^{k+1}, \tag{42}$$

we obtain $T^{k+1}$ via $u^{k+1}$. For completeness, the whole scheme for solving the proposed gradient error prior model with ADMM is shown in Algorithm 1.

**Computational Complexity Analysis.** According to the proposed optimization algorithm 1, and given an input image with size of $m \times n$, we can get the computational complexity as follows. The computational complexity of the fast Fourier transform and inverse fast Fourier transform both are $\mathcal{O}(mn \log(mn))$. The soft-thresholding shrinkage is $\mathcal{O}(mn)$, while the truncated nulcear normal and singular value decomposition are $\mathcal{O}(mn)$ and $\mathcal{O}(mnr)$, and the other subproblems are $\mathcal{O}(mn)$. Therefore, the whole algorithm 1 has a $\mathcal{O}(mnr + mn \log(mn))$ computational complexity.

### A.3 SOLUTION TO GELR WITH NETWORK

First of all, we review the $u$-subproblem, denoted as:

$$u^{k+1} = \arg\min_u ||g - u - T^k||_2^2 + \frac{\rho_1}{2}||\nabla u - d^k + \eta_d^k||_2^2 + \frac{\rho_2}{2}||\nabla u - \nabla x - t^k + \eta_t^k||_2^2. \quad (43)$$

To enable the model to handle multi-scale textures while giving the network better capability of adjusting smooth intensity, we consider $u$-subproblem as a loss function, which drives the proposed network to update and optimize parameters. And we can get $u^{k+1}$ from trained neural network. Therefore, let $u^{k+1} = f_\theta(g)$. Other subproblems would have slight changes, discussed as follows.

**Update $d^{k+1}$** by the soft-thresholding skrinkage, denoted as

$$d^{k+1} = \textbf{shrink}(\nabla f_\theta(g) + \eta_d^k, \frac{1}{\rho_1}). \quad (44)$$

**Update $t^{k+1}$** by $\text{SVT}_{\mu,r}$. Therefore, Let $Q = \nabla f_\theta(g) - \nabla x - t - \frac{\beta}{\rho_2}A^T B + \eta_t^k$, we have

$$t^{k+1} = \arg\min_t \frac{\beta}{\rho_2}||t||_* + \frac{1}{2}||t - Q||_2^2. \quad (45)$$

We have the unique closed-form solution is

$$t^{k+1} = \text{SVT}_{\frac{\beta}{\rho_2},r}(Q). \quad (46)$$

Then, we have

$$t^{k+1} = \text{SVT}_{\frac{\beta}{\rho_2},r}(Q) = U diag[max(\sigma - \frac{\beta}{\rho_2}), 0]V^T, \quad (47)$$

where $U \in \mathbb{R}^{r \times m}$, $V \in \mathbb{R}^{r \times n}$, and $\sigma = (\sigma_1, \cdots, \sigma_r)^T \in \mathbb{R}^r$ are from the Singular Value Decomposition of $Q$.

**Update $\eta_d^{k+1}$ and $\eta_t^{k+1}$** by

$$\begin{cases} \eta_d^{k+1} = \eta_d^k + (\nabla f_\theta(g) - d^{k+1}), \\ \eta_t^{k+1} = \eta_t^k + (\nabla f_\theta(g) - \nabla x - t^{k+1}), \end{cases} \quad (48)$$

the two Lagrange multipliers can be updated directly.

**Update $T^{k+1}$** by

$$T^{k+1} = g - f_\theta(g), \quad (49)$$

In summary, the whole scheme for the gradient error prior guided network model is shown in Algorithm 2.

---

**Algorithm 2** GELR with Network-ADMM

---

**Input:** Image $g$, $x$, $\alpha$ $\beta$, $\rho_1$, $\rho_2$, $\theta$, $K$.
**Initialization:** $T^0, t^0, d^0, \eta_t^0, \eta_d^0 = \mathbf{0}$, $u^0 = g$.
1: **for** $k = 1 : K$ **do**
2:     Update $u^{k+1}$ via the RPAFNet with adam;
3:     Update $d^{k+1}$ by Eq. equation 44;
4:     Update $t^{k+1}$ via Eq. equation 47;
5:     Update $\eta_d^{k+1}$ and $\eta_t^{k+1}$ via Eq. equation 48;
6:     Update $T^{k+1}$ by Eq. equation 49;
7: **end for**
**Output:** $u$.

---

**Computational Complexity Analysis.** Since the algorithmic complexity of the neural network is related to numbers of parameters and layers, and the update of the neural network depends on the GPU, it is meaningless to calculate the algorithmic complexity. According to the proposed optimization algorithm 2, and given an input image with size of $m \times n$, we can get the computational complexity as follows. The soft-thresholding shrinkage is $\mathcal{O}(mn)$, while the truncated nulcear normal and singular value decomposition are $\mathcal{O}(mn)$ and $\mathcal{O}(mnr)$, and the other subproblems are $\mathcal{O}(mn)$. Therefore, the whole algorithm 2 has a $\mathcal{O}(mnr)$ computational complexity.

## A.4 CONVERGENCE ANALYSIS

This section presents the convergence of solving GELR model with designed network. The lemma on the convergence of ADMM and some essential assumptions are from in (Wang et al., 2019b).

First of all, we provide definitions pertinent to the Lipschitz differentiable. For any function $f$ is continuous or differentiable on its domain, we can claim that the function $f$ is Lipschitz differentiable if it is differentibale and its gradient is Lipschitz continuous. Additionally, we need define restricted prox-regularity for regularize objetive functions and an essential Lemma.

**Definition A.2** (**Restricted Prox-Regularity** (Wang et al., 2019b; Hou & Li, 2025))**.** Given a lower semi-continuous function $f\colon \mathbb{R}^n \to \mathbb{R} \cup \{\infty\}$, and $C \in \mathbb{R}_+$, such that

$$S_C := \{u \in dom(f) : ||d|| > C, \forall d \in \partial f(u)\}. \tag{50}$$

$f$ is called restricted prox-regular if $\forall C > 0$ and bounded set $P \subseteq dom(f)$. Then $\exists \gamma > 0$, such that

$$f(g) + \frac{\gamma}{2}||u - g||_2^2 \geq f(u) + <d, g - u>, \tag{51}$$
$$\forall u \in P \backslash S_C, g \in P, d \in \partial f(u), ||d|| \leq C.$$

**Lemma A.3** ((Wang et al., 2019b))**.** *Given the following general optimization problem, denoted as*

$$\arg\min_{X_k, Y} f(X_0, X_1, \cdots, X_p) + h(Y), \quad \text{s.t.} \sum_{k=0}^{p} A_k X_K + BY = C, \tag{52}$$

*where the function $f\colon \mathbb{R}^{n(p+1) \times m} \to \mathbb{R}$ is proper, continous, and possibly nonsmooth. While the function $h\colon \mathbb{R}^{q \times m} \to \mathbb{R}$ is proper and differentiable. $f, h$ can be non-convex. Let $(X^t, Y^t, Z^t)$ be a sequence generated by ADMM framework of equation 52. $Z^t$ is the dual variable, $\mu$ is a psotive parameter. And $\mathcal{L}_\mu$ is the corresponding Lagrangian function. Assume that the following conditions hold:*

*A1(coercivity). Define the feasible set*

$$\mathcal{F} := \left\{ (X, Y) \in \mathbb{R}^{(np+q) \times m} | AX + BY = 0 \right\},$$

*the objective function $f + h$ is corecive over this set, that means $f(X) + h(Y) \to \infty$ if $(X, Y) \in \mathcal{F}$ and $||(X, Y)|| \to \infty$;*

*A2(feasibility). $Im(A) \subseteq Im(B)$, where $Im(\cdot)$ returns the image of a matrix;*

*A3(Lipschitz sub-minimization paths).*

*(1) For any fixed $X$, $\arg\min_Y\{f(X) + h(Y) | BY = U\}$ has a unique minimizer. $H\colon Im(B) \to \mathbb{R}^{q \times m}$ defined by $H(U) := \arg\min_Y\{f(X) + h(Y) | BY = U\}$ is a Lipschitz continuous map.*

*(2) For $k = 0, \cdots, p$, we denote*

$$X_{-k} := (X_0, \cdots, (X_{k-1}, (X_{k+1}, \cdots, (X_p)$$

*and for any fixed $X_{-k}, Y$,*

$$\arg\min_{X_k}\{f(X_k, X_{-k}) + h(Y) | A_k X_k = U\}$$

*has a unique minimizer, and $F_k\colon Im(A_k) \to \mathbb{R}^{p \times m}$ defined by $F_k(U) := \arg\min_{X_k}\{f(X_k, X_{-k}) + h(Y) | A_k X_k = U\}$ is a Lipschitz continuous map.*

*A4(objective-$f$ regularity). $f$ has the form $f(X) = r(X) + \sum_{k=0}^{p} f_k(X_k)$, where $r(X)$ is Lipschitz differentiable with a constant $L_r$, and $f_0(X_0)$ is lower semi-continuous, $f_k(X_k)$ is restricted prox-regular for $k = 1, \cdots, p$;*

*A5(objective-$h$ regularity). $h(Y)$ is Lipschitz differentiable with a constant $L_h$;*

*Specifically, if $\mathcal{L}_\mu$ is a Kurdyka-Łojasiewicz (KŁ) function, then for any sufficiently large $\mu$, $(X^t, Y^t, Z^t)$ converges globally to the unique limit point $(X^*, Y^*, Z^*)$, which satisfies $0 \in \mathcal{L}_\mu(X^*, Y^*, Z^*)$.*

Upon the Lemma as mentined above, To illustrate the convergence, we need to verify that the iterative framework of our proposed algorithm satisfies the **A1-A5** in Lemma A.3 and demonstrates the KŁ property of our augmented Lagrangian function.

**Proof.** Suppose $u$ can be directly obtained via $u = f_\theta(g)$, we rewrite the optimization problem equation 16, denoted as

$$\arg\min_{d,t} \quad \alpha \sum_{i=1}^{2} ||d_i||_1 + \beta \sum_{i=1}^{2} ||t_i||_r, \tag{53}$$

$$\text{s.t.} \quad g = u + T, \quad \nabla_i u = d_i, \quad \nabla_i u - \nabla_i x = t_i.$$

Let $d = [d_1, d_2]$ and $t = [t_1, t_2]$, the corresponding augmented Lagrangian function is

$$\mathcal{L}(u, d_1, d_2, t_1, t_2, \eta_d, \eta_t, T) = \frac{1}{2}||g - u - T||_2^2 + \frac{1}{2}||u - f_\theta(g)||_2^2 + \alpha \sum_{i=1}^{2} ||d_i||_1 \tag{54}$$

$$+ \frac{\rho_1}{2}||\nabla u - d + \eta_d||_2^2 + \beta \sum_{i=1}^{2} ||t_i||_r + \frac{\rho_2}{2}||\nabla u - \nabla x - t + \eta_t||_2^2,$$

Then, we can consider $f(X)$ as $f(X) = \alpha \sum_{i=1}^{2} ||d_i||_1$, and $h(V)$ denotes as

$$h(V) = \beta \sum_{i=1}^{2} ||t_i||_r + \frac{1}{2}||g - u - T||_2^2 + \frac{1}{2}||u - f_\theta(g)||_2^2,$$

where $X = [d_1; d_2]$, and $V = [u; t_1; t_2; T]$. Therefore, we let

$$A = \begin{bmatrix} -I & 0 \\ 0 & -I \\ I & 0 \\ 0 & I \\ 0 & 0 \end{bmatrix} \quad B = \begin{bmatrix} \nabla_1 & 0 & 0 & 0 \\ \nabla_2 & 0 & 0 & 0 \\ 0 & -I & 0 & 0 \\ 0 & 0 & -I & 0 \\ I & 0 & 0 & I \end{bmatrix} \quad C = \begin{bmatrix} 0 \\ \nabla_1 x \\ \nabla_2 x \\ g \end{bmatrix}.$$

Suppose the gradient operators $\nabla_1$ and $\nabla_2$ are with zero boundary condition. Therefore, $B$ has full column rank. In this condition, we can verify that the assumptions **A1-A5** and the KŁ property hold.

The feasible set is $\mathcal{F} = \{(X, V)|AX + BV = C\}$, when $||(X, V)||_2 \to +\infty$, $f(X) + h(V) \to +\infty$. Thus **A1** holds.

Since $\text{Im}(B) = \mathbb{R}^{3mn}$, **A2** naturally holds.

In section A.3, we have presented the unique solution for each subproblem, and $A$, $B$ both have full column rank with trivial null spaces. Then, we can get $F$ and $H$ are linear map operators. Therefore, for any $k_1, k_2 \in \mathbb{N}$, we have

$$||F_i(X^{k_1}) - F_i(X^{k_2})|| \le ||F_i||||X^{k_1} - X^{k_2}||,$$

and

$$||H(V^{k_1}) - H(V^{k_2})|| \le ||H||||V^{k_1} - V^{k_2}||.$$

Thus **A3** holds.

For **A4**, let $r = 0$, $f_0 = 0$ and $f_1 = ||d||_1$, According to Examples in (Poliquin & Rockafellar, 1996). $f_1$ is pro-regular. Therefore, **A4** naturally holds.

For **A5**, we have

$$h(V) = \beta \sum_{i=1}^{2} ||t_i||_r + \frac{1}{2}||g - u - T||_2^2 + \frac{1}{2}||u - f_\theta(g)||_2^2,$$

thus **A5** obviously holds.

For the KŁ property of $\mathcal{L}_\mu$, based on the Example 2 in (Bolte et al., 2014), $\mathcal{L}_\mu$ is a semi-algebraic and it satisfies the KŁ property.

Since the all conditions **A1-A5** hold and $\mathcal{L}_\mu$ meets the KŁ property, for any sufficiently large penalty parameters, the iterative sequence $(u^k, d_1^k, d_2^k, t_1^k, t_2^k, \eta_d^k, \eta_t^k, T^k)$ produced via the proposed GELR model with network converges globally to the unique limit point $(u^*, d_1^*, d_2^*, t_1^*, t_2^*, \eta_d^*, \eta_t^*, T^*)$ and it has $0 \in \partial\mathcal{L}_\mu(u^*, d_1^*, d_2^*, t_1^*, t_2^*, \eta_d^*, \eta_t^*, T^*)$. ∎

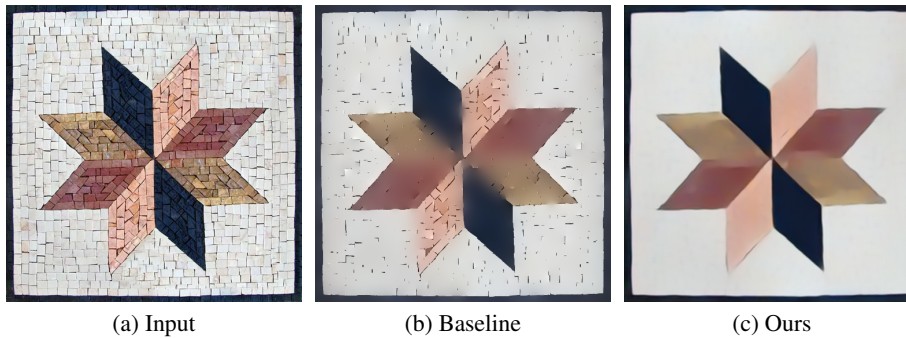

|  (a) Input  |  (b) Baseline  |  (c) Ours  |

Figure 8: Ablation study on loss functions. (a) Input, (b) Baseline means $\lambda_1 = 0$ in equation 15. (c) Full total loss.

## B  ADDITIONAL ABLATION STUDIES

**Visual ablattion results of** $\lambda_1, \lambda_2$. Figure 8 presents the ablation results. As shown in Figure 8(b), the baseline struggles to control smoothing intensity, leading to residual textures and over-smoothed edges. In contrast, Figure 8(c) shows that our GELR model, optimized via ADMM, effectively adjusts the smoothing level, producing cleaner and more structurally consistent results. Furthermore, GELR enable RPAFNet to flexibly balance smoothing and detail preservation. We have reported different values of $\lambda_1, \lambda_2$ impact on filtering performance in Table 5. The best choice of $\lambda_1, \lambda_2$ is 0.7, 0.3. We present more choice of parameters $\lambda 1, \lambda 2$ and $\alpha, \beta$ verse the filtering performance in Figure 9. The value of $\lambda_1$ is larger, the greater the smoothing strength. The value of $\lambda_2$ is larger, the more structural and edge information is kept. Additionally, the value of $\alpha$ is larger, the greater the smoothing strength. The value of $\beta$ is larger, the more structural and edge information is remained. Empirically, to achieve a better tradeoff between smoothing strength and edge information preservation, we set $\alpha = 0.4, \beta = 0.6$ in this study.

Table 5: Quantitative results of different values for $\lambda_1, \lambda_2$.

| $(\lambda_1, \lambda_2)$ | (0.1,0.9) | (0.2,0.8) | (0.3,0.7) | (0.4,0.6) | (0.5,0.5) |
|---|---|---|---|---|---|
| **BRISQUE↓** | 50.265 | 48.960 | 45.045 | 42.571 | 39.540 |
| **NIQE ↓** | 9.472 | 8.352 | 7.827 | 7.012 | 6.846 |
| $(\lambda_1, \lambda_2)$ | (0.6,0.4) | (0.7,0.3) | (0.8,0.2) | (0.9,0.1) | (1.0,0) |
| **BRISQUE↓** | 34.254 | **30.041** | 36.415 | 40.176 | 45.251 |
| **NIQE↓** | 6.512 | **5.972** | 6.480 | 7.091 | 8.273 |

**CTUM ablation results**. It enables the fusion of both local and global representations, thereby enriching the feature space. This design allows the model to better reconstruct fine details while maintaining global coherence, resulting in more visually detailed outputs. We have reported quantitative results of the ablation experiment for the CTUM, as shown in Table 6. We present the visual results of the ablation study on the CTUM model in Figure 10. It can be seen that Figure 10 (b) is the same neural network without deploying the CTUM model, and this subfigure shows its loss of structural information and edges in the smoothed image. In contrast, the full neural network with the CTUM produces a higher-quality filtering output, keeping more structures and edges. These results well illustrate the contribution of the CTUM in image filtering tasks.

Table 6: Quantitative results of ablation study on CTUM module.

| Networks | Metrics | BRISQUE↓ | PIQE↓ | NIQE ↓ |
|---|---|---|---|---|
| Baseline | w/o | 28.537 | 51.698 | 8.631 |
| RPAFNet | w | **21.665** | **30.275** | **4.307** |

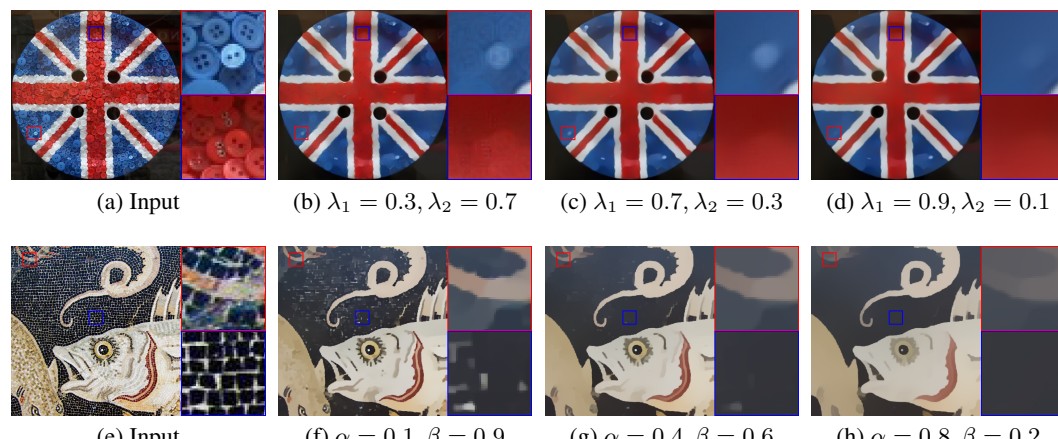

(a) Input     (b) $\lambda_1 = 0.3, \lambda_2 = 0.7$     (c) $\lambda_1 = 0.7, \lambda_2 = 0.3$     (d) $\lambda_1 = 0.9, \lambda_2 = 0.1$

(e) Input     (f) $\alpha = 0.1, \beta = 0.9$     (g) $\alpha = 0.4, \beta = 0.6$     (h) $\alpha = 0.8, \beta = 0.2$

Figure 9: Visual effects of parameters $\lambda_1, \lambda_2$ and $\alpha, \beta$. It is evident that the value of $\lambda_1$ is larger, the greater the smoothing strength. The value of $\lambda_2$ is larger, the more structural and edge information is kept. Additionally, the value of $\alpha$ is larger, the greater the smoothing strength. The value of $\beta$ is larger, the more structural and edge information is remained.

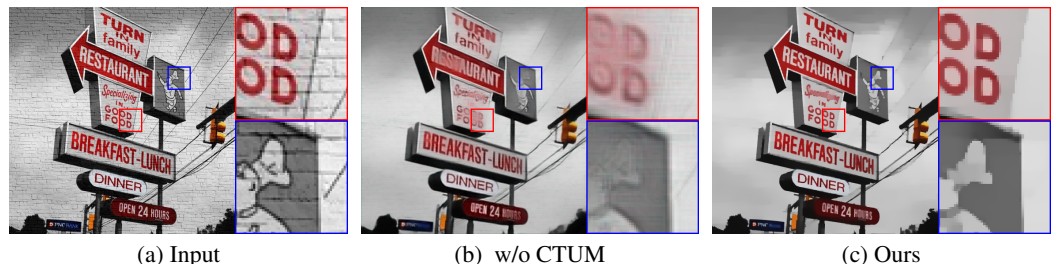

(a) Input        (b) w/o CTUM        (c) Ours

Figure 10: Visual effects for CTUM ablation study. (a) Input, (b) Without deploying the CTUM module, (c) Our with deploying the CTUM module. It is worth noting that the CTUM helps to keep more structures and edges.

**Dilation rate ablation results**. The dilation rate is a parameter in the LDSC module. The choice of dilation rate is based on the experimental visual effects. Therefore, we conduct an ablation study on the dilation rate as shown in Figure 11. The large dilation rate benefits in suppressing large textures, but causes over-smoothing in structure and edges. We can observe this phenomenon in Figure 11. Therefore, it is better to choose a smaller dilation rate $\{1, 2, 4\}$ for preserving structures and edges.

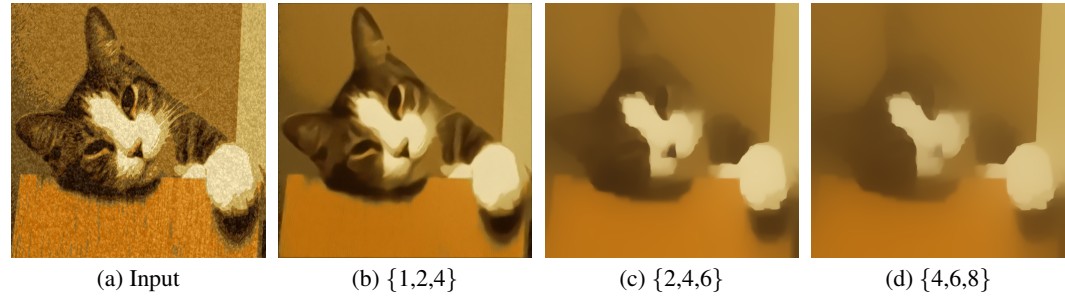

(a) Input        (b) $\{1,2,4\}$        (c) $\{2,4,6\}$        (d) $\{4,6,8\}$

Figure 11: Visual effects of ablation study on dilation rate. It is clear that a large dilation rate causes an over-smoothing risk in the output images.

**Downsampling ratio ablation study**. The downsample ratio is a hyperparameter in our RPAFNet. To illustrate its impact on the filtering performance, we conduct an ablation study on the downsample ratio in Figure 12. We present four different choices for downsample ratios, corresponding to $\{0.4, 0.6, 0.8, 1.0\}$. It can be seen that the ratio is 0.6, which obtains competitive results with that of 0.8. Overall, the ratio is set as 0.8 to keep more structural information and avoid oversmoothing. It is worth noting that the ratio is set as 1.0 means there is no downsampling operator, which makes it hard to remove large textures completely. Therefore, we set the downsample ratio to 0.8 for all experiments in this study.

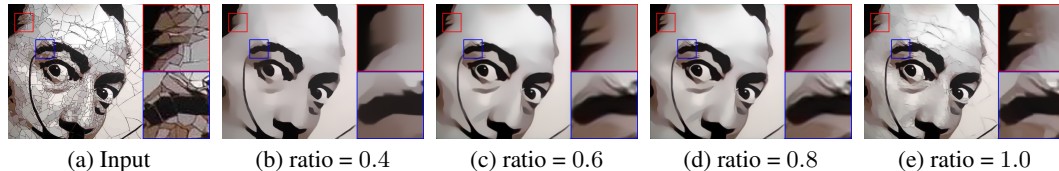

| (a) Input | (b) ratio = 0.4 | (c) ratio = 0.6 | (d) ratio = 0.8 | (e) ratio = 1.0 |

Figure 12: Downsampling ratio ablation experimental results. It is evident that a smaller downsampling ratio causes oversmoothing, while a larger downsampling ratio suffers from incomplete texture filtering.

**Ablation study on GELR**. To verify the effect of the GELR, we compare the performance of the GELR implementation in the non-deep framework with that of GELR applied in the proposed network. The visual comparison of the GELR is shown in Figure 13. For the large-scale texture scenario, GELR can filter small textures, while it is unable to handle large and irregular textures. For the small texture of nature, GELR obtained competitive results over the proposed model. These results demonstrate that introducing GELR to a neural network can improve its ability to handle long-range textures.

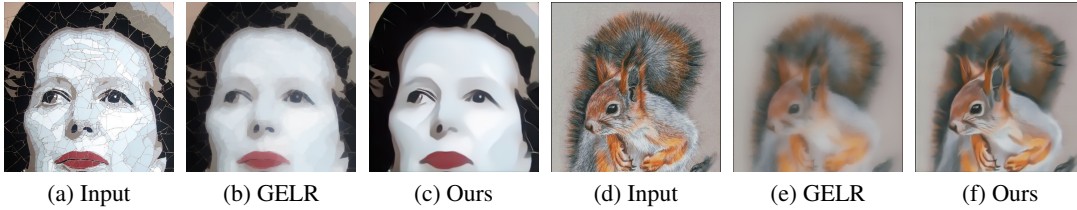

| (a) Input | (b) GELR | (c) Ours | (d) Input | (e) GELR | (f) Ours |

Figure 13: Visual comparison for the GELR model. The GIRL model can effectively remove small textures, while being limited in handling large-scale textures, as shown in (b) and (e). In contrast, integrating GELR into the neural network can improve the ability to handle large-scale textures, as shown in (c).

**Ablation study on SSIM loss term**. To verify that the proposed model does not introduce metric bias, we report the performance of a version trained without the SSIM loss. We present the visual effects of the SSIM ablation study in Figure 14. Plot (a) is the input texture image, corresponding to the clean image (b). Plot (c) denotes the output of the trained network without the SSIM loss. In contrast, Plot (d) is the output of the proposed model with the SSIM loss. It can be seen that deploying the SSIM yields a slight improvement in structural and edge preservation, as evidenced by the PSNR and SSIM values, which illustrate this conclusion. However, this improvement did not result in a dramatic performance gap. Therefore, we can almost conclude that there is no issue of artificially inflating performance due to obvious metric bias.

## C  APPLICATION EXPERIMENTS

To further illustrate the proposed model's performance, we utilize three downstream tasks to compare our approach against the state-of-the-art methods, across three smoothing applications, which include details manipulation, image stylization and clipart compression artifact filtering.

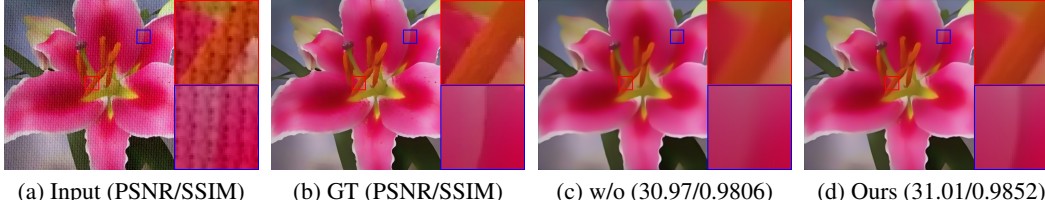

| (a) Input (PSNR/SSIM) | (b) GT (PSNR/SSIM) | (c) w/o (30.97/0.9806) | (d) Ours (31.01/0.9852) |

Figure 14: Visual effects of the SSIM term ablation study. (a) Input, (b) GT, (c) Without the SSIM loss term, (d) Ours with the SSIM loss term. It can be seen that deploying the SSIM yields a slight improvement in structural and edge preservation. The minimal gap in PSNR and SSIM values illustrates that there is no significant metric bias in RPAFNet.

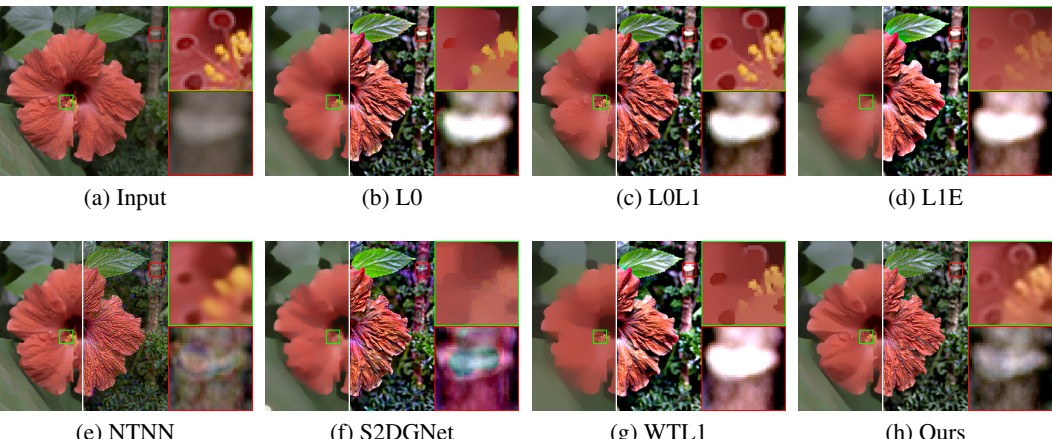

| (a) Input | (b) L0 | (c) L0L1 | (d) L1E |
| (e) NTNN | (f) S2DGNet | (g) WTL1 | (h) Ours |

Figure 15: Details Manipulation. (a) The input image. It is enhanced with four details layers via SOTA methods: (b) L0, (c) L0L1, (d) L1E, (e) NTNN, (f) S2DGNet, (g) WTL1, (h) Ours. The right green enlarged area denotes the details of smoothed image, and the red-marked enlarged boxes are corresponding enhanced details. From these marked areas, one can see that our method can reduce halo artifacts. Meanwhile, it keeps more edges than other algorithms.

## C.1 DETAILS MANIPULATION.

It enhances details by incorporating multiple texture layers, extracted by subtracting the smoothed image from the original. We present details manipulation results of different SOTA techniques in Figure 15. The left part of each image is the smoothed image, while the right parts are corresponding detail enhanced images. Green enlarged areas show details from smoothed results and red enlarged areas reveal details from enhanced images. L0, L1E and S2DGNet oversmoothed edges in green marked boxes. L0L1, NTNN, and WTL1 obtained competive edges. However, L0, L0L1, L1E and WTL1 suffer white halo artifacts, and also NTNN, S2DGNet produce colorful halo artifacts. In contrast, the proposed model obtains the best visual effect, which reduces significant halo artifacts.

To directly evaluate the performance of reducing gradient reversal artifacts in image enhancement, we compare the proposed model with other filtering approaches in Figure 16. It is evident that imRTV, RTV, and Deepwls have significant gradient reversal artifacts, as shown in the red-marked and enlarged regions. NTNN obtains a slight gradient reversal artifact. In contrast, our method has the best visual effects in removing gradient reversal artifacts.

## C.2 IMAGE STYLIZATION.

This task aims to transform an input image into an image with new style, while preserving the main contents. This technique can abstract the content of low-contrast areas while preserving the high-contrast features of images. Stylization results of the comparison approaches are shown in Figure

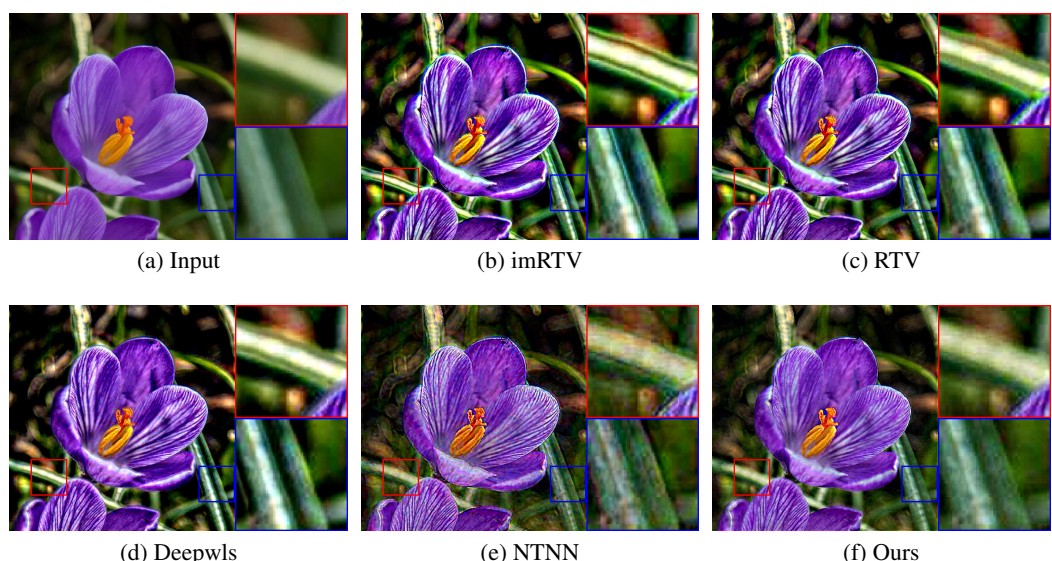

Figure 16: Gradient reversal artifacts removal. (a) Input, which is enhanced with $4\times$ details layers by (b) imRTV, (c) RTV, (d) Deepwls, (e) NTNN, and (f) Ours. It can be seen that the proposed model obtains superiority in reducing gradient reversal artifacts over the compared approaches.

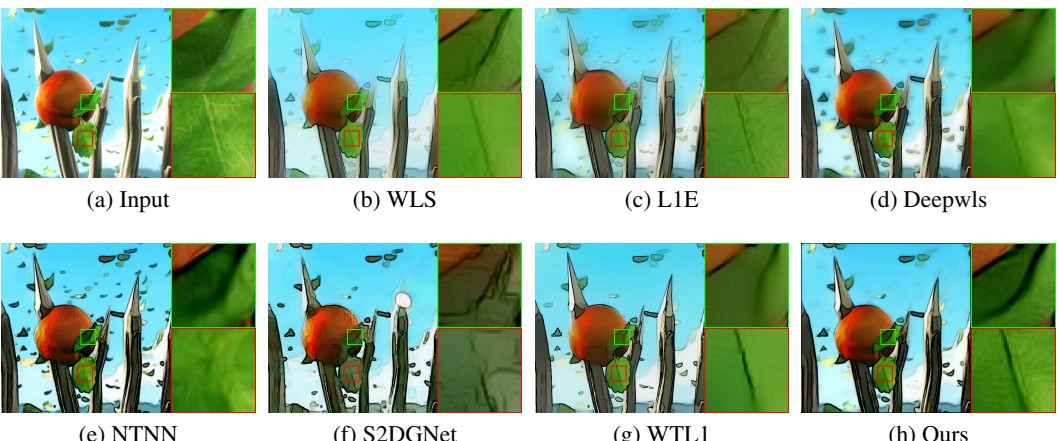

Figure 17: Image stylization. (a) The input image, stylization results of (b) WLS, (c) L1E, (d) Deepwls, (e) NTNN, (f) S2DGNet, (g) WTL1, (h) Ours. From these marked and enlarged areas, one can see that our model has significant advantages in the main structures preserving.

17. It is worth noting that WLS, L1E, Deepwls, and NTNN can not preserve the high-contrast edges, leading to oversmoothing. We recommend focusing on these green and red highlighted areas. S2DGNet and WTL1 obtain competitive performance, while they also can not do the best in emphasizing high-contrast structures. By contrast, the proposed model has a significant superiority in structure keeping and obtains the best visual effects over the compared techniques.

## C.3 ARTIFACTS FILTERING.

The technique of compression artifacts filtering aims to be employed to eliminate JPEG block artifacts when converting clip-art images into JPEG format. When an image is compressed at a low bit rate using standard JPEG encoding, compression artifacts often manifest along sharp edges, while staircase artifacts may arise in homogeneous regions. In this study, a 10% compression rate is ap-

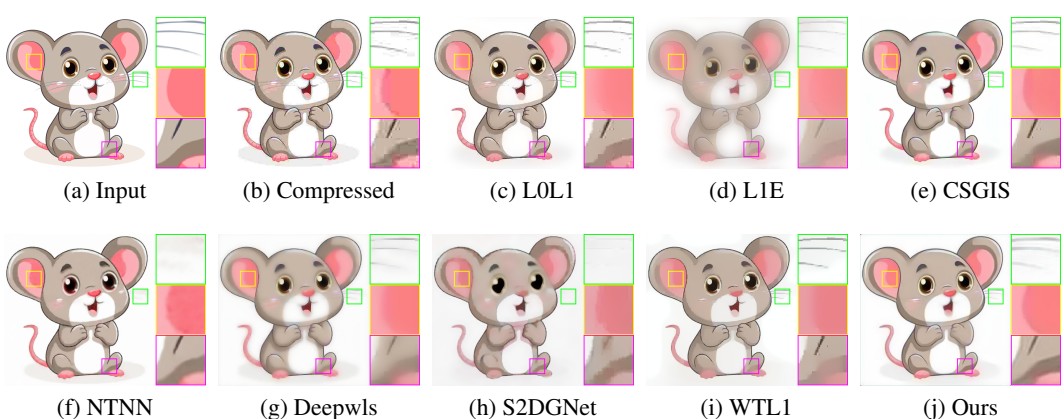

(a) Input  (b) Compressed  (c) L0L1  (d) L1E  (e) CSGIS

(f) NTNN  (g) Deepwls  (h) S2DGNet  (i) WTL1  (j) Ours

Figure 18: Clip-arts JPEG artifacts filtering. (a) Given input image, (b) The compressed image. It is smoothed by (c) L0L1, (d) L1E, (e) CSGIS, (f)NTNN, (g) Deepwls, (h) S2DGNet, (i) WTL1, and (j) Ours. Referring to the marked boxes, (c), (d), (f), (g) and (h) suffer from blurred edges. (e) and (i) filter artifacts uncleanly.

Table 7: PSNR (dB) and SSIM comparison for artifacts filtering.

| Methods | L0L1 | L1E | CSGIS | Deepwls |
|---------|------|-----|-------|---------|
| PSNR | 27.86 | 20.63 | 27.03 | 24.46 |
| SSIM | 0.8931 | 0.8460 | 0.8912 | 0.8889 |
| Methods | NTNN | S2DGNet | WTL1 | Ours |
| PSNR | 27.88 | 22.81 | 24.66 | **28.67** |
| SSIM | 0.8935 | 0.8640 | 0.8122 | **0.9151** |

plied to the given clip-art image. Figure 18 illustrates the smoothed results obtained from various SOTA models. It is evident that L1E, Deepwls blur the input image. L0L1, CSGIS, and WTL1 filter the JPEG blocks uncleanly. Meanwhile, L0L1 and S2DGNet also produce staircase edges. NTNN and S2DGNet oversmoothed the details of the input image, referring to contents of the green enlarged boxes. However, the proposed method keeps better edges and details while removing JPEG block artifacts cleanly. We also show the numerical PSNR and SSIM values corresponding to this task in Table 7. Our proposed model demonstrates the best performance, achieving the highest PSNR value of 28.67 and an SSIM value of 0.9151. Whatever the visual effects or quantitative numerical metrics, the proposed model achieves the best performance against other SOTA methods in the removal of compression artifacts.

To verify that the proposed model does not produce gridding artifacts, we present four FFT graphs in Figure 19, which are obtained by our model. From the corresponding FFT graphs, it is evident that there are no significant gridding artifacts in the smoothed outputs. These results illustrate that the adoption of multiple dialtion combinations has no risk of gridding artifacts in RPAFNet.

## C.4 EXPERIMENTS ON PUBLIC DATASETS

We conduct smoothing experiments across on the three public datasets, including SPS (Feng et al., 2021), NKS (Xu et al., 2020), and ECS (Qi et al., 2024), three above datasets have paired ground-truth smoothed images. The smoothed images are shown in Figure 20. The first row of images are from NKS dataset, the second row of images are from SPS dataset, and the last row of images are from ECS dataset. For these results of NKS dataset, L1E, Deepwls, and WTL1 suffer from blurring and over-smooothing to varying degrees. S2DGNet obtains competitive filtering results. For these ouputs of SPS dataset, the all compared techniques filter textures uncleanly. For these smoothed images of ECS dataset, It is evident that L1E, Deepwls blur output images, while S2DGNet and WTL1 over-smoothing details. In contrast, the proposed model obtain the best visual effect across three datasets, demonstrated the robustness of our technique. The corresponding numerical values

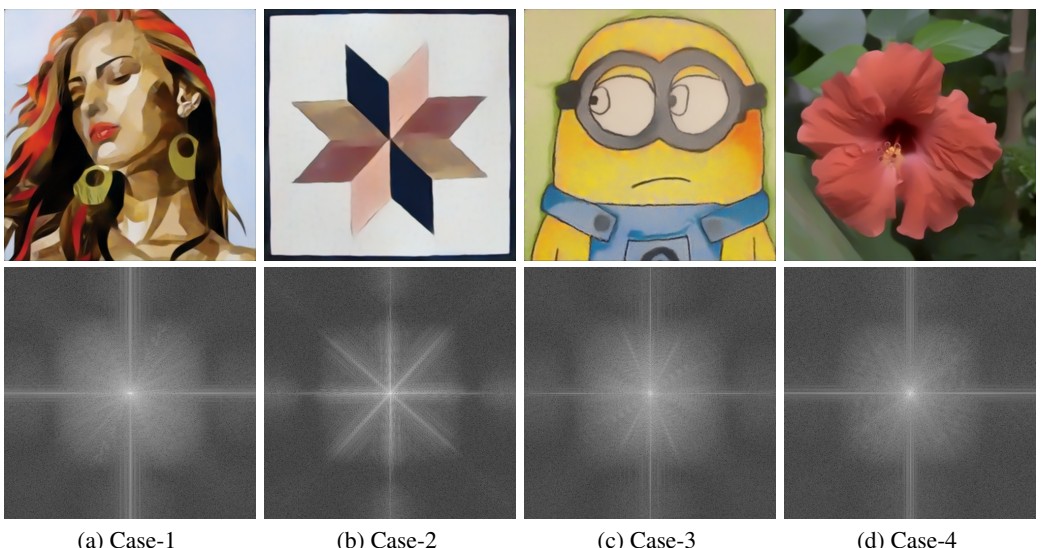

(a) Case-1          (b) Case-2          (c) Case-3          (d) Case-4

Figure 19: Visual effects of FFT graphs. The first row denotes the smoothed images, obtained from the proposed RPAFNet, the second row denotes the corresponding FFT images. it is evident that there are no significant gridding artifacts in the smoothed outputs of the RPAFNet.

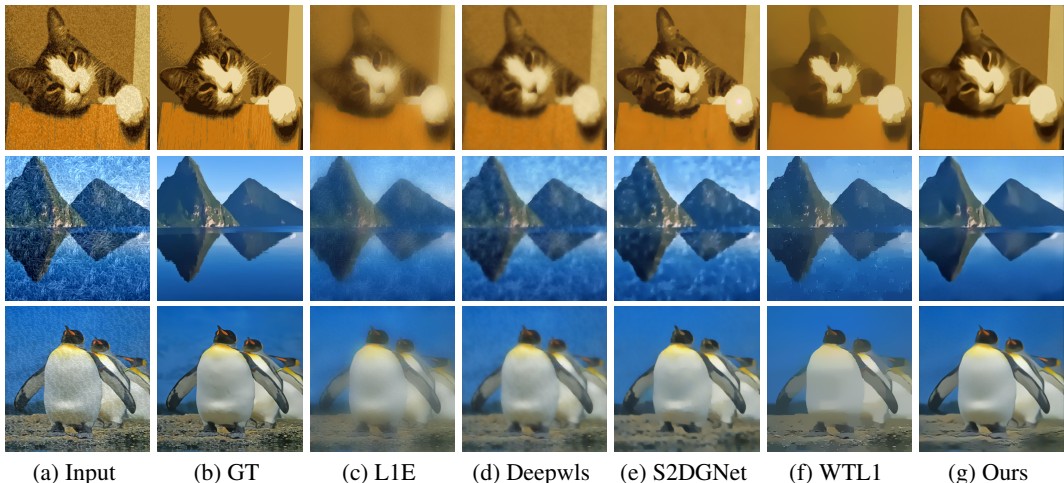

(a) Input     (b) GT     (c) L1E     (d) Deepwls     (e) S2DGNet     (f) WTL1     (g) Ours

Figure 20: Image smoothed results across three datasets. (a) The input images, filtered by (c) L1E, (d) Deepwls, (e) S2DGNet, (f) WTL1, (g) Ours, and corresponding their (b) GT. The first row of images are from NKS dataset, the second row of images are from SPS dataset, and the last row of images are from ECS dataset. It is evident that L1E, Deepwls blur output images, while Deepwls, S2DGNet and WTL1 filter textures uncleanly. In contrast, the proposed model obtain the best visual effect across three datasets, demonstrated the robustness of our technique.

of PSNR and SSIM are presented in Table 8. It is evident that our model achieves the best index in smoothing on three public datasets.

## C.5 TASK-SPECIFIC MODEL COMPARISON

To verify the performance of the proposed model, we tested it on multiple downstream tasks using smoothing techniques. Downstream tasks include image detail enhancement, image stylization, and reducing compression artifacts. It is worth noting that our model performs no better than these

Table 8: PSNR(dB) and SSIM values across three public datasets.

| Methods | Datasets | | | | | |
|---|---|---|---|---|---|---|
| | NKS | | SPS | | ECS | |
| | PSNR | SSIM | PSNR | SSIM | PSNR | SSIM |
| L0 (2011) | 27.36 | 0.8932 | 27.85 | 0.8826 | 25.37 | 0.8539 |
| L0L1 (2022) | 25.60 | 0.8425 | 26.21 | 0.8633 | 24.58 | 0.7946 |
| L1E (2022) | 26.15 | 0.8651 | 24.68 | 0.8295 | 25.14 | 0.8021 |
| ILS (2020) | 25.98 | 0.8863 | 25.57 | 0.8413 | 23.25 | 0.7488 |
| QWLS (2024) | 28.27 | 0.8924 | 27.57 | 0.8739 | 25.16 | 0.8049 |
| SEMF (2023) | 28.46 | 0.8962 | 27.71 | 0.8892 | 24.75 | 0.7983 |
| WLS (2008) | 23.56 | 0.8014 | 24.56 | 0.8541 | 22.89 | 0.8014 |
| CSGIS (2022) | 34.50 | 0.9486 | 24.56 | 0.8541 | 24.90 | 0.8701 |
| E2H (2021) | 34.24 | 0.9401 | 31.73 | 0.9202 | 26.89 | 0.8914 |
| Deepwls (2023) | 27.63 | 0.8876 | 26.55 | 0.8798 | 24.87 | 0.8153 |
| NTNN (2024) | 29.89 | 0.9035 | 28.57 | 0.9024 | 26.49 | 0.8849 |
| S2DGNet (2024) | 33.76 | 0.9503 | 32.15 | 0.9302 | 30.45 | 0.9101 |
| WTL1 (2024) | 25.43 | 0.8519 | 26.47 | 0.8718 | 24.76 | 0.7395 |
| Ours | **34.98** | **0.9575** | **32.68** | **0.9382** | **31.68** | **0.9286** |

Table 9: Quantitative results of task-specific model comparsion.

| Methods | DAGN | EDAR | Ours |
|---|---|---|---|
| PSNR | 31.41 | 29.34 | 28.53 |
| SSIM | 0.8954 | 0.8080 | 0.7942 |

task-specific methods in downstream tasks, since we have no paired images for downstream tasks to train our model. To present the performance gap between the proposed method and these task-specific models, we utilize the compression artifacts removal task to show this conclusion in Figure 21. Plot (a) exhibits compression artifacts, obtained by compressing the clean image (b) with a 10% compression bitrate. It is evident that DAGN and EDAR have significant superiority over the proposed model in this task. We report the corresponding PSNR and SSIM values in Table 9. DAGN and EDAR models obtain higher index values than that of the proposed model.

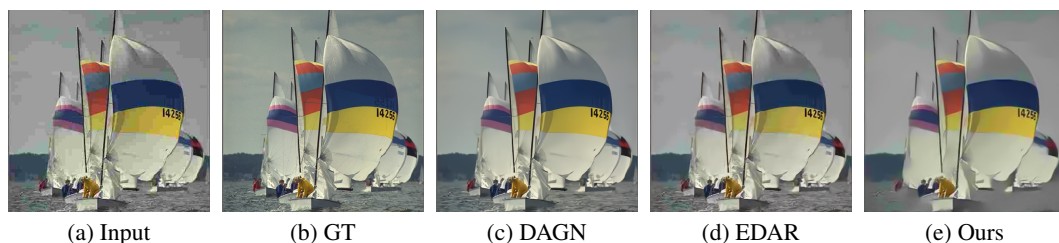

| (a) Input | (b) GT | (c) DAGN | (d) EDAR | (e) Ours |

Figure 21: Task-specific model comparison. (a) Input with compression artifacts, (b) GT, results of (c) DAGN, (d) EDAR, and (e) Ours. It is worth noting that our model performs no better than these task-specific methods in compression artifacts removal task. DAGN and EDAR have kept more details, while the proposed model over-filtering details and structural information.

## C.6 FAILURE CASES ANALYSIS

We have discussed the limitations of the proposed model in Section 5. One of the limitations is that it is hard to handle low-contrast textures. Therefore, we present two failure cases of the proposed model in Figure 22. Our model is unable to preserve those structures and edges in low-contrast texture regions, as referred to the red and blue marked areas. We plan to design a specific mathematical model or add a network module to solve this problem in future work.

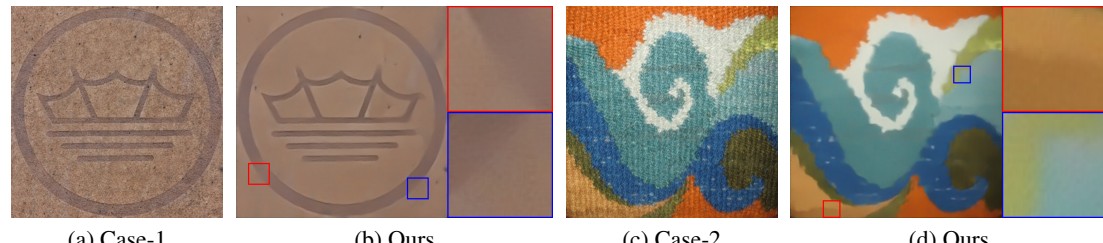

|          (a) Case-1          |          (b) Ours          |          (c) Case-2          |          (d) Ours          |

Figure 22: Failure cases on the low-contrast scenarios. From the enlarged areas, it can be seen that the proposed model is unable to preserve those structures and edges in low-contrast texture regions.

### C.7 FUTURE METHODOLOGY EXTENSION

This study is designed for the image smoothing tasks. However, the proposed GELR can indeed be interpreted as a general structural prior that emphasizes edge preservation and texture suppression. This type of prior is closely related to the denoising objectives in score-based diffusion models and to structure-consistency constraints used in generative pipelines.

GELR can be inserted during the sampling process, which is similar to plug-and-play priors, serving as a structure-preserving constraint that suppresses over-generated textures. Meanwhile, GELR can be added as a structural regularizer during score matching to improve the fidelity of structural components and reduce hallucinated textures. These directions highlight that our method can be extended to be applied to other computer tasks. We will further explore these directions in our subsequent studies.

### C.8 INFERENCE PERFORMANCE

To present the inference performance of the proposed model, we leverage two images with sizes of 512x512 and 2048x2048 to conduct this experiment. Quantitative results are shown in Table 10. Results are obtained using PyTorch on a Ubuntu 20.04 server with two RTX 4090 GPUs. The latency time of the proposed model is less than other methods.

Table 10: Inference performance comparsion.

| Methods | Model size | Latency | GPU Memory Usage |
|---------|------------|---------|------------------|
|         |            | $512 \times 512$ / 2K | |
| Deepwsl | 0.3M | 0.18s/0.34s | 164.5M/2609.9M |
| NTNN | 0.57M | 0.32s/0.67s | 193.7M/3112.9M |
| S2DGNet | 4.28M | 1.28s/3.51s | 376.5M/5471.3M |
| WTL1 | 11.04M | 0.19s/0.24s | 200.5M/2492.1M |
| Ours | 16.07M | 0.15s/0.19s | 859.7M/3384.5M |

### C.9 STRUCTURAL EVALUATION

There are no general objective metrics to measure texture-scale uniformity directly. Currently, we only confirm the texture-scale via humans. RTV (Xu et al., 2012) provides a public smoothing dataset with paired edge images. Therefore, to conduct a structural evaluation, we use PSNR and SSIM metrics to compare the performance of different approaches in edge preservation. These edge images are obtained by the Canny operator. The corresponding average PSNR and SSIM values are shown in Table 11. It is evident that our model obtain the best index values, which means the proposed model can maintain more structure and edges.

Table 11: Quantitative results of Structural Evaluation.

| Methods | Deepwls | NTNN | WTL1 | L1E | S2DGNet | Ours |
|---------|---------|------|------|-----|---------|------|
| PSNR | 28.75 | 29.46 | 26.83 | 27.49 | 29.15 | **30.25** |
| SSIM | 0.8519 | 0.8944 | 0.8835 | 0.8457 | 0.9027 | **0.9172** |

