# OpenReview forum: "Residual Pyramid Atrous Filtering Network with the Error Low-Rank Respresentation"
_ICLR.cc/2026/Conference — Submitted to ICLR 2026_

### Official Review · Reviewer_9XUC · 2025-10-27

**Soundness:** 3
**Presentation:** 2
**Contribution:** 2
**Rating:** 6
**Confidence:** 3

**Summary:**

The paper proposes RPAFNet with two structure modifications (LDSC and DRL) and one theoretical model (GELR) to address the challenge in image filtering, that is, 1) balancing multi-scale texture suppression and 2) structural edge preservation. Based on the LDSC and DRL, the RPAFNet is capable of establishing multi-range correlation and preserving edge information. Additionally, the GELR uses $||\nabla u||_1$  regularization to avoid edge blurring and suppress over-smoothing and $\beta||\nabla u-\nabla x|| $ to precisely reduce textures through low-rank approximation. Overall, the RPAFNet achieves remarkable results in several experiments.
​

**Strengths:**

- The motivation is clear, and the overall story is sound. The RPAFNet is based on both theoretical (GELR) and engineering optimization (LDSC, DRL).
- The theoretical derivation is solid.
- The experimental results are stable to show the effectiveness of the proposed model.

**Weaknesses:**

- Despite the proposed method being able to extend to multiple downstream tasks,  the comparison is conducted on generic methods like Deepwls and WTL1. Since it is an NN-based model without performance advantages, it is important to show the performance gap with the task-specific model.
- The inference performance and comparison are not included, and the training overhead for introducing GELR.
- The theoretical analysis is clear; however, it might be more persuasive to provide a visual comparison for the proposed GELR.
- The LSDC and DRL are not novel modules but a modification of existing methods. The design of dilation convolution (large kernel convolution) is widely used and explored.

**Questions:**

See weakness.

---

> ### Author Response · Authors · 2025-11-20
> **Response for the Reviewer 9XUC**
>
> We greatly appreciate Reviewer 9XUC for the positive assessment and constructive feedback. We address each question as follows:
>
> ---
>
> **[Q1] Regarding comparison with task-specific model:**
>
> We sincerely appreciate your insightful questions and constructive feedback. We want to clarify that the primary focus of this paper is image filtering, and the paired training data is special for the smoothing task. We use **downstream tasks as indirect evaluations to demonstrate the performance of different smoothing techniques**. Importantly, **our model is not trained on any downstream-task-specific paired data**.  For example, in the image enhancement or JPEG artifact removal tasks, we directly apply the smoothing model—trained solely on the smoothing dataset—to these tasks during inference, without any fine-tuning or re-training. In contrast, the task-specific models are trained on paired data tailored for those downstream tasks. Therefore, **a direct performance comparison is inherently unfair and naturally favors those task-specific models**.
>
> Nevertheless, **to illustrate the performance gap and provide a more complete understanding, we still include both visual and quantitative comparisons with two task-specific models (DAGN [1] and EDAR [2]) on JPEG artifact removal (compression rate 10%)**. These **results are provided in Figure 21 of Appendix C.5**.  DAGN and EDAR obtain superiority over the proposed model. Specifically, **DAGN and EDAR maintain more details than our model in Figure 15, and their PSNR and SSIM values are higher than ours**, which is shown in the table below.
>
> | Methods | DAGN | EDAR | Ours |
> |:---:|:---:|:---:|:---:|
> | PSNR |31.41|29.34|28.53|
> | SSIM |0.8954|0.8080| 0.7942|
>
> ---
>
> [1] Ma, Li, Yifan Zhao, Peixi Peng, and Yonghong Tian. "Sensitivity decouple learning for image compression artifacts reduction." IEEE Transactions on Image Processing 33 (2024): 3620-3633.
>
> [2] Yu, Ke, Chao Dong, Chen Change Loy, and Xiaoou Tang. "Deep Convolution Networks for Compression Artifacts Reduction." CoRR (2016).
>
> ---
>
> **[Q2] Regarding inference performance:**
>
> We appreciate your valuable questions. We leverage **two images with sizes of 512x512 and 2048x2048 to compare inference performance**. Quantitative results are shown in the table below. Results are obtained using PyTorch on a Ubuntu 20.04 server with two RTX 4090 GPUs.
>
> | Methods | Model size | Latency | GPU Memory Usage |
> |:---:|:---:|:---:|:---:|
> ||| **512x512** / **2K** |  |
> | Deepwsl |0.3M|0.18s/0.34s|164.5M/2609.9M  |
> | NTNN |0.57M|0.32s/0.67s| 193.7M/3112.9M |
> | S2DGNet|4.28M|1.28s/3.51s|376.5M/5471.3M|
> | WTL1 |11.04M|0.19s/0.24s|200.5M/2492.1M|
> | Ours |16.07M|0.12s/0.16s|859.7M/3384.5M|
>
> ---
>
> Since the GELR covers the SVD operator,  the training overhead for introducing GELR and the GPU memory has been increased by 186MB for a **batchsize of 4 and an image size of $512\times 512\times 3$**.
>
> ---
>
> **[Q3] Regarding visual comparison for GELR:**
>
> We greatly thank you for raising this important point. Firstly, we appreciate your positive feedback on our theoretical analysis. Following your suggestion, **we have added a visual comparison of GELR, which is shown in Figure 13** of the revised manuscript. We can draw that **GELR alone struggles to handle multi-scale textures**, particularly those with large spatial variation. To address this limitation, we integrate GELR into our RPAFNet, which provides multi-scale feature extraction and enables the model to better capture textures at different scales. We also observe that GELR performs similarly to our method on fine-scale textures, which confirms the ability of our model to handle large-scale textures.
>
> ---
>
> **[Q4] Regarding the LSDC and DRL modules:**
>
> We greatly appreciate your insightful comments. The LDSC module is built on the well-known atrous convolutions. We have well cited the related paper in the manuscript. We acknowledge that the dilation convolution is widely used and explored. However, it is precisely **the success of large kernel convolution** in extracting features from **large textures** that we consider as a unit to integrate into the LSDC module. The LDSC module is designed to have three branches, which can **effectively extract different-scale features**, thereby enhancing its capacity to **feature representations**. The DRL module extracts **differences of features** between the previous layer and the skip layer. We aim to leverage the DRL module to **compensate for structural information** during the decoding stage, reducing the loss of edges and dominant structures.
>
> While the changes introduced in the LSDC and DRL modules are **incremental rather than fundamentally innovative**, experimental results demonstrate that these refinements lead to improved smoothing performance.
>
> ---
>
> We have revised the relevant contents in the manuscript. Thank you again for your contributions in helping us improve our paper.
>
> ---

---

> > ### Comment · Reviewer_9XUC · 2025-11-28
> >
> > Thanks for the authors' detailed rebuttal and explanation. The response addresses my current concerns about inference and training performance. Despite still being concerned with the novelty of LSDC and DRL,  I would keep my positive score.

---

### Official Review · Reviewer_Wi8X · 2025-10-28

**Soundness:** 2
**Presentation:** 2
**Contribution:** 2
**Rating:** 2
**Confidence:** 4

**Summary:**

This paper presents a network called RPAFNet (Residual Pyramid Atrous Filtering Network) for image smoothing, aiming to suppress textures while preserving fine structural details. The model adopts a typical U-shaped encoder–decoder architecture, enhanced with two modules: LDSC (Large-Dilation Spatial Convolution) to expand the receptive field, and DRL (Difference Residual Layer) to strengthen feature reconstruction. In addition, the authors propose a Gradient Error Low-Rank Representation (GELR) that combines Total Variation (TV) with a low-rank constraint. Using an ADMM-based optimization strategy, the GELR term is incorporated as an additional loss to further improve structural preservation during smoothing.

**Strengths:**

1.Well-organized and technically sound

The paper is clearly structured, with a well-defined model design and mathematically consistent derivations. The experiments are reasonably comprehensive, showing good implementation quality and logical consistency.

2.Potential for further improvement and extension

Although the novelty is limited, the work provides a clear and systematic framework for traditional image smoothing models. It also leaves room for future extensions, such as incorporating differentiable optimization or generative modeling methods (e.g., DPO or diffusion/flow-based models).

**Weaknesses:**

1.Limited architectural and module-level innovation

RPAFNet adopts a conventional U-shaped encoder–decoder framework without introducing substantial structural novelty. The LDSC module relies on atrous convolution to expand the receptive field, a technique that was thoroughly explored in the DeepLab series (2017). In contrast, modern approaches typically employ Transformer or hybrid attention mechanisms to handle long-range dependencies, making the proposed design appear outdated. Meanwhile, the DRL module performs structure compensation merely through feature differencing, showing strong similarity to standard residual or skip connections. It lacks clear theoretical justification and empirical validation of its independent contribution. Overall, the design appears to be an integration of existing techniques rather than a breakthrough in architectural paradigm.

2.Conservative loss design and potential optimization conflicts.

The loss formulation mainly follows the traditional “TV + low-rank” joint regularization framework, which has been extensively used in earlier image restoration and smoothing tasks. The paper introduces no novel constraint or optimization mechanism. Furthermore, the simultaneous use of structural loss L1, perceptual loss L2, and an SSIM term in both training and evaluation may cause gradient conflicts and metric bias. Since SSIM is explicitly optimized during training, its improvement in evaluation may partially result from direct loss fitting rather than genuine structural enhancement.

3.Lack of explanation for experimental settings

The paper employs bilinear interpolation with a down sampling ratio of 0.8, which is an unconventional choice. However, no clear motivation, empirical justification, or performance sensitivity analysis is provided, limiting the interpretability and reproducibility of the results.

**Questions:**

1.On the architectural design and innovation

The paper repeatedly highlights the “limited receptive field” as a key bottleneck, but this issue has already been addressed in recent architectures such as Transformer or hybrid attention networks. Could the authors clarify how the LDSC module provides advantages over standard atrous convolution, multi-scale structures, or Transformer-based architectures in terms of receptive field and feature representation? It would be helpful to include quantitative or qualitative comparisons to demonstrate its unique contribution.
Furthermore, the DRL module performs structure compensation mainly through feature differencing, which appears conceptually similar to residual or skip connections. Could the authors elaborate on the motivation and theoretical foundation of DRL, and provide ablation results to justify its necessity and effectiveness compared with simpler alternatives such as residual fusion or attention-based refinement?

2.On loss design and optimization objectives

The current “TV + Low-rank” joint regularization follows a traditional formulation without introducing new constraints or optimization mechanisms. Moreover, combining (L_1) and (L_2) losses may introduce conflicting gradient directions and affect structure preservation, have the authors observed such instability?
Additionally, since the SSIM term is used in both training and evaluation, could this cause metric bias and artificially inflate performance? It would strengthen the validity of the results if the authors could report outcomes of a version trained without SSIM loss, or analyze its impact empirically.


3.On experimental settings

The paper uses bilinear interpolation with a down sampling ratio of 0.8, which is quite unusual. The authors should clarify during the rebuttal why this value was chosen — for example, to ensure gradient smoothness, multi-scale feature consistency, or based on empirical tuning — and analyze its sensitivity to this parameter.

4.On methodological extension and potential research value

While the paper focuses on image smoothing, it conceptually overlaps with texture suppression and structure enhancement problems often addressed by generative models. Have the authors considered applying the proposed framework in generative or diffusion-based architectures (e.g., Stable Diffusion, Flow Matching) to evaluate its scalability and potential cross-domain benefit? Such discussion would help clarify the method’s broader applicability and originality.

---

> ### Author Response · Authors · 2025-11-20
> **Response for the Reviewer Wi8X**
>
> We greatly appreciate Reviewer Wi8X for the positive assessment and constructive feedback. We address each question as follows:
>
> ---
>
> **[Q1] Regarding limited architectural and module-level innovation:**
>
> We sincerely thank the reviewer for the thoughtful comments. We agree that RPAFNet adopts a U-shaped encoder–decoder backbone and that atrous convolution is not new by itself. However, **the key novelty of our method does not lie in proposing an entirely new architectural paradigm, but rather in task-specific design choices that integrate gradient-domain priors with dilation convolutional mechanisms for image smoothing**.
>
> Regarding the **LDSC module**. It utilizes atrous convolution operators with different dilation rates for handling long-range dependencies in filtering tasks, which is more lightweight than  Transformer or hybrid attention mechanisms.
>
> Regarding the **DRL module**. DRL is not a simple residual skip connection, which acts as a structural information compensation between different levels and scales. It extracts the differences from the previous layer and the skip layer. Subsequently, combining these differences into the decoder used for reconstruction.
>
> The **novelty of the proposed RPACNet is contributed by LDSC, CTUM, DRL, and the GELR optimization model together**. Despite not being a new paradigm in architecture, this integration provides substantial performance improvement, which we believe is valuable for low-level vision tasks. **More details are shown in [Q4].**
>
> ---
>
> **[Q2] Regarding Conservative loss design:**
>
> We thank the reviewer for the insightful comments. Our goal is not to introduce a new optimization paradigm, but to **bridge classical structure–texture priors with neural training**. Although TV and low-rank priors are well studied, previous works apply them as explicit optimization constraints on the output images. In contrast, we leverage the TV regularizer to constrain the output directly, while the low-rank regularizer is applied on the gradient error features. It is worth noting that **it is significantly different from earlier image restoration and smoothing tasks.**
>
> Regarding the **optimization conflicts**. The loss is not a simple combination, and **each term serves a distinct role**. $L_{1}$ enforces local structural fidelity to sharp edges. The low-rank corresponding loss is designed to suppress textures. SSIM loss ensures the structural similarity between the output and the GT directly. Meanwhile, we leverage $\alpha,\beta$ and $\lambda_{1}, \lambda_{2}$ to balance each term.
>
> Empirically, removing any component leads to noticeable filtering quality degradation (as shown in the ablation studies of Appendix B ), showing that the loss design has no conflicts. **More details are shown in [Q5].**
>
> ---
>
> **[Q3] Regarding the bilinear interpolation:**
>
> We greatly appreciate the reviewer’s comments. We acknowledge that bilinear interpolation with a downsampling ratio of 0.8 is an unconventional choice. Meanwhile, this is not the novelty and contribution of the proposed network.  **The choice of the 0.8 downsampling ratio follows the commonly adopted experimental protocol in prior literature [1,2,3]**. We intentionally align with these works to ensure fair comparison, reproducibility, and consistency of the evaluation pipeline. Meanwhile, **we provide an ablation study on the downsampling ratio in the revised manuscript. More details are shown in [Q6].**
>
> [1] Qing Zhang, Hao Jiang, Yongwei Nie, and Wei-Shi Zheng. 2023. "Pyramid Texture Filtering." ACM Transactions on Graphics 42, 4, (2023), 11.
>
> [2] Jiang, Hao, Rongjia Zheng, Yongwei Nie, Chunxia Xiao, Wei-Shi Zheng, and Qing Zhang. "Self-supervised Texture Filtering." ACM Transactions on Graphics 44, no. 5 (2025): 1-13.
>
> [3] Yang, Yang, Dan Wu, Lanling Zeng, and Zhuoran Li. "Weighted least square filter via deep unsupervised learning." Multimedia Tools and Applications 83, no. 11 (2024): 31361-31377.
>
> ---

---

> > ### Author Response · Authors · 2025-11-20
> >
> > **[Q4] Regarding architectural design and innovation:**
> >
> > We sincerely thank the reviewer for the thoughtful comments and fully agree that receptive-field modeling and structural compensation are central to the filtering task. We acknowledge that the success of Transformer or hybrid attention networks lies in modeling long-range dependencies. Actually,  we also applied the Transformer in our CTUM module for the decoder.
> >
> > **Regarding the advantage of the LDSC module**. LDSC is built upon dilated convolution; its design is tailored specifically for texture smoothing, which behaves very differently from semantic vision tasks. **Unlike DeepLab-style atrous blocks, LDSC intentionally removes BatchNorm and activation (as shown in Fig. 2(b)).**  LDSC uses a minimal set of dilation rates (1, 2, 4) and a 1×1 fusion layer to provide **a stable, lightweight receptive field expansion without amplifying high-frequency errors**. Meanwhile, the usage of dilated convolution has fewer parameters than the Transformer-based module. Finally, we respectfully highlight that our ablation study (Fig. 6, Table 2) directly demonstrates that LDSC alone reduces multi-scale textures significantly and obtains better BRISQUE and PIQE scores, supporting its unique contribution.
> >
> > **Regarding the advantage of the DRL module**. DRL explicitly computes the feature difference between encoder feature $f_e$ and corresponding decoder feature $f_d$:
> > $$\Delta f = f_d-f_e$$
> > followed by an $L_2$-normalization to stabilize magnitude (as shown in Fig. 2(e)). **This difference map highlights structural edges—which are scale-inconsistent—while suppressing repetitive textures, which aligns directly with the edge-preserving objective in Eq.(3)**. In contrast, conventional skip connections pass absolute features and do not isolate structural components. Residual blocks perform within-layer refinement, not cross-scale structural compensation. The necessity of DRL is empirically validated in our ablation study (Fig. 7): removing DRL drops PSNR from 27.43 to 25.25 and SSIM from 0.9065 to 0.8529. We greatly appreciate the reviewer’s concerns. **LDSC and DRL were designed specifically for the challenges of texture smoothing, and both ablations consistently show substantial improvements**.
> >
> > ---
> >
> > **[Q5] Regarding loss design and optimization objectives:**
> >
> > We sincerely appreciate the reviewer’s thoughtful comments on the loss formulation. These concerns regarding the classical nature of “TV + low-rank” regularization, the potential interaction between $L_1$, $L_2$, and SSIM terms, and the possibility of SSIM-related metric bias are all very reasonable. We would like to clarify that the main novelty lies in how these priors are used. The proposed GELR model introduces a non-convex gradient-error low-rank representation and couples it with the network through an ADMM-guided dynamic constraint.
> >
> > **Regarding gradient conflict**, we have not observed such instability in the output images. The reasons can be attributed to that $L_1$ (from GELR), $L_2$, and SSIM play complementary roles. $L_1$ enforces gradient consistency and sharp edges; $L_2$ maintains pixel fidelity; SSIM enhances structural coherence with the ground-truth. **We leverage parameters $\alpha,\beta$ and $\lambda_{1}, \lambda_{2}$ to balance each term (ablation studies are shown in the ablation studies of Appendix B)**. Additionally, the low-rank term mainly regularizes on the gradient error map rather than the gradient of the output image directly, resulting in no existing gradient conflicts.
> >
> > **Regarding SSIM term**, in our all experiment, SSIM denotes as below:
> > $$SSIM(f_{\theta}(g),x) = 1-ssim(f_{\theta}(g),x).$$
> >
> > $ssim$ is the structural similarity index. We have clarified the SSIM description in the revised manuscript. **To validate the SSIM metric bias, we have added an ablation study on it, as shown in Figure 14**. We find that without the SSIM loss causes performance has a slight degradation, which confirms that improvements are not due to metric overfitting.
> >
> > ---

---

> > > ### Author Response · Authors · 2025-11-20
> > >
> > > **[Q6] Regarding experimental settings:**
> > >
> > > We sincerely thank the reviewer for highlighting the downsampling ratio issue. We fully agree that using a ratio of 0.8 may appear unusual compared with more standard values, and we appreciate the opportunity to clarify our motivation. Empirically, we found that overly aggressive downsampling causes noticeable structural over-smoothing, while very mild downsampling results in inadequate filtering of multi-scale texture and weaker smoothing capability. **The choice of the 0.8 downsampling ratio follows the commonly adopted experimental protocol in prior literature [1,2,3]**.  However, following the reviewer’s suggestion, **we conducted ablation studies on downsampling ratios with ${0.4, 0.6, 0.8, 1.0}$; the corresponding results consistently indicated that 0.8 produces the better filtering performance and better quantitative metrics, as shown in Figure 12 of the revised manuscript**.
> > >
> > > In summary, although 0.8 may appear unconventional at first glance, it is the outcome of empirical tuning and a design choice. We appreciate the reviewer for raising this point, and we provide a sensitivity analysis in **Appendix B**.
> > >
> > > [1] Qing Zhang, Hao Jiang, Yongwei Nie, and Wei-Shi Zheng. 2023. "Pyramid Texture Filtering." ACM Transactions on Graphics 42, 4, (2023), 11.
> > >
> > > [2] Jiang, Hao, Rongjia Zheng, Yongwei Nie, Chunxia Xiao, Wei-Shi Zheng, and Qing Zhang. "Self-supervised Texture Filtering." ACM Transactions on Graphics 44, no. 5 (2025): 1-13.
> > >
> > > [3] Yang, Yang, Dan Wu, Lanling Zeng, and Zhuoran Li. "Weighted least square filter via deep unsupervised learning." Multimedia Tools and Applications 83, no. 11 (2024): 31361-31377.
> > >
> > > ---
> > >
> > > **[Q7] Regarding methodological extension and potential research value:**
> > >
> > > We greatly appreciate the reviewer’s thoughtful suggestions. We fully agree that image smoothing, texture suppression, and structure enhancement share conceptual similarities with the priors used in modern generative models, especially diffusion-based frameworks. The reviewer’s comment provides an important perspective on the broader applicability of our method.
> > >
> > > Our work is designed for the image smoothing tasks. However,  the proposed Gradient Error Low-Rank Representation (GELR) can indeed be interpreted as a general structural prior that emphasizes edge preservation and texture suppression. This type of prior is closely related to the denoising objectives in score-based diffusion models and to structure-consistency constraints used in generative pipelines.
> > >
> > > We have carefully discussed and analyzed how our framework may be incorporated into modern generative models. Several extensions exist:
> > >
> > > (1) **Plug-in prior for diffusion sampling.** GELR can be inserted during the sampling process (similar to plug-and-play priors), serving as a structure-preserving constraint that suppresses over-generated textures.
> > >
> > > (2) **As a regularizer for score-based networks.** GELR can be added as a structural regularizer during score matching to improve the fidelity of structural components and reduce hallucinated textures.
> > >
> > > (3) **Applied for cross-domain guidance.** GELR provides explicit control over texture removal; it can be used as a controllable operator within Flow Matching or Stable Diffusion pipelines for structure-aware generation or style abstraction.
> > >
> > > These directions highlight that our method can be extended to be applied to other computer tasks. We include a dedicated discussion paragraph in the revised manuscript to clarify the broader applicability of GELR and outline the above extensions. In fact, the idea raised by the reviewer opens up a valuable avenue for future research, and we view it as a compelling follow-up topic that could significantly broaden the impact of our approach. We will further explore this direction in our subsequent studies.
> > >
> > > ---
> > >
> > > We have clarified these points in the revised manuscript. Thank you again for your contributions in helping us improve our paper.
> > >
> > > ---

---

> > > > ### Comment · Reviewer_Wi8X · 2025-11-27
> > > >
> > > > ## Response to Q1 in the first-round review (regarding the proposed LDSC module)
> > > >
> > > > In my initial comment, I pointed out that, given the widespread adoption of Transformer or hybrid attention mechanisms in modern methods to model long-range dependencies, the proposed LDSC module appears relatively traditional in its design. In their response, the authors emphasized that LDSC is “more lightweight than Transformer or hybrid attention mechanisms” and argued that it can achieve stable receptive field enlargement with fewer parameters. While this provides a preliminary explanation from the perspective of lightweight design, it does not adequately address my core concern—whether the module achieves a meaningful trade-off between computational cost and modeling capability. The authors did not provide a systematic comparison between LDSC and mainstream long-range dependency modeling approaches in terms of efficiency and performance, nor did they demonstrate clear advantages of LDSC in practical scenarios. Therefore, I believe the current response is insufficient to support the claim that LDSC is a justified or advanced core module design.
> > > >
> > > > ## Response to Q1 in the first-round review (regarding the proposed DRL module)
> > > >
> > > > The authors conducted ablation studies to show that adding the DRL module improves performance, which confirms its usefulness. However, the current version only shows that "adding this module helps," but do not explain DRL with alternative designs (e.g., residual fusion, attention-based refinement). Therefore, it remains unclear whether the performance gain comes from the module's design or simply from added complexity.
> > > >
> > > > ## Response to Q2 in the first-round review (regarding evaluation metrics)
> > > >
> > > > The authors justified the use of SSIM in the training loss. However, continuing to use SSIM as the main evaluation metric raises concerns, as the model was explicitly optimized for it. This may lead to overfitting to the metric itself rather than reflecting genuine improvements in perceptual or structural quality.

---

### Official Review · Reviewer_QPdk · 2025-10-28

**Soundness:** 2
**Presentation:** 2
**Contribution:** 2
**Rating:** 6
**Confidence:** 3

**Summary:**

The paper proposes an image filtering network called RPAFNet, which aims to remove multi-scale textures while preserving edges. The network consists of two novel core modules:  Lightweight Dilated Spatial Convolution (LDSC) module and Difference Residual Layer (DRL). Lightweight dilated convolution is used to extract multi-scale texture features, and the difference between encoded features is used as a skip connection, emphasizing high-frequency information. An additional Gradient Error Low Rank (GELR) model is introduced to calculate a non convex optimization term based on low rank approximation at the loss end to further suppress texture and preserve edges.

**Strengths:**

1. The optimization is novel. Monotonically decreasing iterative closed form solutions for the GELR objective function is provided and convergence to a limit point is proved, efficiently suppressing texture and overcoming oversmoothing issues.

2. Integrating "dilated pyramid feature extraction+differential residual skip connection+low rank texture suppression" into an end-to-end framework, the three complement each other and have a clear idea.

3. The method is demonstrated with rich experiments. The quantitative indicators are comprehensively leading and the qualitative results are impressive.

**Weaknesses:**

1. LDSC uses multiple dilation combinations, whick might induce gridding artifacts. It’s suggested that using FFT graph to verify whether this risk exists in RPAFNet.

2. Theoretical assumption is too strong. GELR convergence proof relies on the assumption of "Lipschitz continuity and bounded gradient", and whether the actual network feature map distribution satisfies this assumption has not been verified. Suggest providing statistics on the gradient/Lipschitz constant during the training process.

3. Sensitivity curves were not performed for the combination of dilation rate and low rank rank value r.

**Questions:**

1. What’s $T^k$ in Equation 13?

2. Has the combination of different dilation rates been systematically searched? Is there a better "receptive field scheduling" strategy (such as dynamic dilation)?

3. Is the design friendly to high-resolution images (such as 4K)? Does the computational complexity increase with the square of resolution?

---

> ### Author Response · Authors · 2025-11-20
> **Response for the Reviewer QPdk**
>
> We appreciate Reviewer QPdk for the detailed feedback and thoughtful questions. It greatly helps us to improve our manuscript. We respectfully address each concern below:
>
> ---
>
> **[Q1] Regarding the gridding artifacts:**
>
> We appreciate your insightful suggestions. To demonstrate that no risk of gridding artifacts exists in RPAFNet, **we present FFT graphs of four different images in Figure 19** of the revised manuscript. From these FFT graphs, it is evident that **no significant gridding artifacts** are present in the output images.
>
> ---
>
> **[Q2] Regarding the gradient/Lipschitz constant:**
>
> We appreciate your valuable questions. We agree that the Lipschitz continuity and bounded gradient assumption are key to the convergence analysis in non-convex optimization problems. Specifically, our filtering method can be considered an inner and outer two-layer framework, where the outer is optimized via the alternating direction method of multipliers, and the $u$-subproblem of the inner layer is optimized via the Adam optimizer of PyTorch. **Therefore, to satisfy theoretical assumptions, we need to restrict the input and output of the neural network**. In this study, we **normalize the input and output image pixel values to the range [0, 1]**. Meanwhile, the proposed network is also equipped with a ReLU activation function.  In this way, **the gradient is bounded in the range [-1, 1]**. On the other hand, these assumptions are standard and appear in recent convergence analyses of deep models [1,2], plug-and-play framework [3], and diffusion-based optimization [4]. Our theoretical analysis is based on these prior works.
>
> [1] Luo, Yisi, Xile Zhao, Kai Ye, and Deyu Meng. "Neurtv: Total variation on the neural domain." SIAM Journal on Imaging Sciences 18, no. 2 (2025): 1101-1140.
>
> [2] Shang, Wanqing, Guojun Liu, Yazhen Wang, Jianjun Wang, and Yuemei Ma. "A non-convex low-rank image decomposition model via unsupervised network." Signal Processing 223 (2024): 109572.
>
> [3] Hou, Ruizhi, and Fang Li. "Hyperspectral image denoising via cooperated self-supervised CNN transform and nonconvex regularization." Neurocomputing 616 (2025): 128912.
>
> [4] Rui, Xiangyu, Xiangyong Cao, Li Pang, Zeyu Zhu, Zongsheng Yue, and Deyu Meng. "Unsupervised hyperspectral pansharpening via low-rank diffusion model." Information Fusion 107 (2024): 102325.
>
> ---
>
> **[Q3] Regarding the low rank value $r$:**
>
> Thanks for your insightful questions. The rank $r$ value of our algorithm **is not a manually fixed hyperparameter**. Instead, it is adaptively determined from each input image. Specifically, for an input image, we **perform singular value decomposition (SVD)** and sort the singular values in descending order. Subsequently, we choose **the smallest rank $r$ such that the cumulative energy of the top-$r$ singular values exceeds 80% of the total energy** (refer to [5]).
>
> That means **the rank is data-dependent**, and it varies across different images, allowing the model to automatically capture the intrinsic low-rank structure of each input rather than relying on a heuristic constant rank. This adaptive strategy improves both robustness and generalization.
>
> [5] Chen, Peng, Fang Li, Deliang Wei, and Changhong Lu. "Low-Rank and Deep Plug-and-Play Priors for Missing Traffic Data Imputation." IEEE Transactions on Intelligent Transportation Systems (2024).
>
> ---

---

> > ### Author Response · Authors · 2025-11-20
> >
> > **[Q4] Regarding $T^{k}$ in Eq.(13):**
> >
> > We greatly appreciate your suggestions. **$T$ denotes the texture layer**, which is used to subtract the output from the input $g$. Therefore, **$T^{k}$ is the texture layer corresponding to the $k$-th timestep**.
> >
> > ---
> >
> > **[Q5] Regarding different dilation rates:**
> >
> > We thank the reviewer for raising this important point. **We have not systematically searched** for the combination of different dilation rates. The adopted dilation-rate configuration was selected based on **a simple trial** over a few combinations (e.g., {1,2,4}, {2,4,6}, {4,6,8}), and we observed that a **larger dilation rate chosen causes edge information loss, as shown in Figure 11** of the revised manuscript. Therefore, we set the dilation rate as  {1,2,4} in our experiments.
> >
> > Regarding receptive-field scheduling strategies (e.g., dynamic dilation), these **adaptive mechanisms can potentially enhance representational flexibility**. We did not discuss them in this study.  However, it is worth noting that **exploring the effects of dynamic dilation convolution on filtering performance is a fascinating topic** in the field of image restoration. **The adaptive/dynamic dilation is a promising direction for future investigation**.
> >
> > ---
> >
> > **[Q6] Regarding the computational complexity:**
> >
> > Thanks for your insightful questions. The **computational and memory costs** of our model **increase as the resolution** of the input image rises. We acknowledge that the current model is not ideal for high-resolution inputs. Moreover, the computational cost increases approximately quadratically with image resolution, primarily due to the **Singular Value Decomposition (SVD) operation** in our framework, whose **complexity rises substantially as the spatial dimensions grow**.
> >
> > ---
> >
> > We have clarified these points in the revised manuscript. Thank you again for your contributions in helping us improve our manuscript.
> >
> > ---

---

### Official Review · Reviewer_deHs · 2025-11-04

**Soundness:** 3
**Presentation:** 3
**Contribution:** 3
**Rating:** 4
**Confidence:** 4

**Summary:**

1. Proposed RPAFNet, a residual pyramid atrous filtering network for image smoothing.
2. Introduced LDSC module to extract multi-scale texture features using dilated convolutions.
3. Designed DRL module to enhance feature space via difference residual connections.
4. Developed CTUM module to fuse local and global features for better reconstruction.
5. Formulated a novel GELR model using gradient error low-rank representation for texture suppression.
6. Provided complete theoretical derivation and convergence proof for the optimization algorithm.
7. Demonstrated superior performance over state-of-the-art methods across multiple datasets and applications.

**Strengths:**

1. The paper demonstrates high originality by formulating a novel non-convex optimization model (GELR) that creatively combines a classical total variation term with a low-rank constraint on the gradient error.
2. It presents a significant architectural innovation with its RPAFNet, which integrates purpose-built modules like LDSC and DRL specifically designed to handle multi-scale textures and enrich the feature space.
3. The work is of exceptional quality due to its theoretical rigor, providing complete derivations for the non-convex optimization and a comprehensive convergence analysis for the proposed ADMM algorithm.
4. The experimental quality is thorough and convincing, featuring extensive comparisons against 14 state-of-the-art methods across multiple datasets and meaningful downstream applications.

**Weaknesses:**

1. The GELR model's reliance on ground-truth gradients (∇x) during training severely limits its real-world applicability, restricting it to synthetic datasets and preventing use on natural images where ideal smoothed targets are unavailable.
2. Computational efficiency is completely unanalyzed, with no reporting of inference speed, model size, or comparison to efficient alternatives, making practical utility impossible to assess.
3. The central claim of superior multi-scale texture handling lacks quantitative validation, relying solely on visual examples rather than objective metrics measuring texture-scale uniformity or structural preservation.
4. Critical ablation studies are incomplete, with the CTUM module's impact shown only numerically without visual proof, and the hyperparameter selection process lacking principled justification for balancing texture removal versus edge preservation.
5. Specific artifact analysis is insufficient, failing to provide direct visual evidence of improvement on stated problems like gradient reversal and offering only superficial treatment of the acknowledged low-contrast texture limitation.

**Questions:**

1. The GELR model's reliance on ground-truth gradients (∇x) during training severely limits its real-world applicability, restricting it to synthetic datasets and preventing use on natural images where ideal smoothed targets are unavailable.
2. Computational efficiency is completely unanalyzed, with no reporting of inference speed, model size, or comparison to efficient alternatives, making practical utility impossible to assess.
3. The central claim of superior multi-scale texture handling lacks quantitative validation, relying solely on visual examples rather than objective metrics measuring texture-scale uniformity or structural preservation.
4. Critical ablation studies are incomplete, with the CTUM module's impact shown only numerically without visual proof, and the hyperparameter selection process lacking principled justification for balancing texture removal versus edge preservation.
5. Specific artifact analysis is insufficient, failing to provide direct visual evidence of improvement on stated problems like gradient reversal and offering only superficial treatment of the acknowledged low-contrast texture limitation.

---

> ### Author Response · Authors · 2025-11-20
> **Response for the Reviewer deHs**
>
> We sincerely thank Reviewer deHs for the careful review and insightful questions. We respectfully address each concern below:
>
> ---
>
> **[Q1] Regarding real-world applicability:**
>
> We greatly appreciate your insightful comment and fully acknowledge the limitation. As correctly pointed out, the dependence on paired supervision is a **common limitation** shared by supervised deep frameworks, and our model is not an exception. The gradient ($\nabla x$) supervision used in GELR is mainly a training-time structural constraint, and its **effectiveness depends** on the availability of **paired targets**. Regarding the real-world applicability of our model on natural images,  it can construct **equivariant structural constraints** via traditional filters. Alternatively, it can utilize a small amount of real data for **fine-tuning when deploying the proposed model in real-world scenarios**. Such adaptation is effective in bridging synthetic-to-real gaps in various computer vision tasks.
>
> If this manuscript is accepted, we will extend the discussion of this limitation in the journal version of this study. The proposed filtering technique can naturally be utilized as a backbone in self-supervised or weakly supervised settings or integrated into a few-shot learning paradigm. These directions would eliminate the need for precise gradient targets while preserving the benefits of GELR, and we believe they are great future research topics.
>
> ---
>
> **[Q2] Regarding computational efficiency analysis:**
>
> We appreciate your insightful questions. We leverage **two images with sizes of 512x512 and 2048x2048 to compare inference performance**. Quantitative results are shown in the table below. Results are obtained using PyTorch on a Ubuntu 20.04 server with two RTX 4090 GPUs.
>
> | Methods | Model size | Latency | GPU Memory Usage |
> |:---:|:---:|:---:|:---:|
> ||| **512x512** / **2K** |  |
> | Deepwsl |0.3M|0.18s/0.34s|164.5M/2609.9M  |
> | NTNN |0.57M|0.32s/0.67s| 193.7M/3112.9M |
> | S2DGNet|4.28M|1.28s/3.51s|376.5M/5471.3M|
> | WTL1 |11.04M|0.19s/0.24s|200.5M/2492.1M|
> | Ours |16.07M|0.15s/0.19s|859.7M/3384.5M|
>
> We have added this content to **Appendix C.8** in the revised manuscript and highlighted them.
>
> ---
>
> **[Q3] Regarding the objective metrics:**
>
> We greatly appreciate your suggestions, which help us improve the completeness of our manuscript.  **It is worth noting that there are no general objective metrics to measure texture-scale uniformity directly**. Currently, we only confirm the texture-scale via humans. Meanwhile, **it is an exceptionally promising avenue to design general objective metrics for measuring texture-scale uniformity**. However, to show quantitative validation on structural and edge preservation, we use **PSNR and SSIM** metrics to compare the performance of different approaches in edge preservation. RTV [1] provides a **public smoothing dataset, which maintains paired edge images**. We compare edge images of different filtering techniques; **these edge images are obtained by the Canny operator**. The corresponding average PSNR and SSIM values are shown in the table below.
>
> | Methods | Deepwls | NTNN | WTL1 | L1E | S2DGNet | Ours |
> |:---:|:---:|:---:|:---:|:---:|:---:|:---:|
> |PSNR|28.75 |29.46 |26.83 |27.49 |29.15|**30.25** |
> |SSIM|0.8519|0.8944| 0.8835| 0.8457 | 0.9027 |**0.9172** |
>
> We have add this table to **Appendix C.9** in the revised manuscript, and present corresponding description.
>
> [1] Xu, Li, Qiong Yan, Yang Xia, and Jiaya Jia. "Structure extraction from texture via relative total variation." ACM transactions on graphics (TOG) 31, no. 6 (2012): 1-10.
>
> ---
>
> **[Q4] Regarding ablattion studies:**
>
> We sincerely thank your insightful suggestions. We have added the **visual results of the CTUM module's impact in Figure 10**. Meanwhile, we also present the visual results of **the hyperparameter selection process for balancing texture removal versus edge preservation in Figure 9**. The detailed analysis of these visual results is presented in **Appendix B** of the revised manuscript.
>
> ---
>
> **[Q5] Regarding the gradient reversal:**
>
> We appreciate your valuable questions and thoughtful feedback, which help us improve the quality of our manuscript. We provide **visual effects of reducing the gradient reversal artifacts in Figure 16** of the revised manuscript. From the marked and enlarged regions, it can be seen that the proposed model obtains superiority in reducing gradient reversal artifacts over competing approaches.
>
> Regarding the **low-contrast texture limitation**, we present two **failure cases of our model in Figure 22 of Appendix C.6**. It is worth noting that the proposed model fails to preserve edges in these cases.
>
> ---
> We have clarified these points in the revised manuscript. Thank you again for your contributions in helping us improve our paper.

---

> ### Author Response · Authors · 2025-11-28
> **Response to the Reviewer deHs**
>
> Thank you once again for your thoughtful review and valuable comments! With only seven days remaining before the end of the Discussion Phase, we would greatly appreciate your feedback on our revised responses. Please let us know whether our clarifications adequately address your concerns. If any issues remain unresolved or if further explanation is needed, we would be grateful for your additional guidance.
>
> Best regards, ﻿
>
> Authors

---

### Author Response · Authors · 2025-12-03
**Response Summary**

We sincerely appreciate all reviewers' insightful and thoughtful suggestions. We are pleased that the reviewers recognized our method as **highly original** (Reviewer deHs), **impressive performance** (Reviewer QPdk), **well-organized and technically sound** (Reviewer Wi8X), and **solid theoretical derivation** (Reviewer 9XUC).

Our main responses are as following six folds:

1. Provided **inference performance** comparison in the tables below;

2. Conducted detailed **ablation studies** in Figures 10-14;

3. Clarified the **design reasonableness** of the proposed model and loss functions;

4. Provided more **comprehensive novelty illustration** for the LDSC and DRL modules in RPAFNet;

5. Clarified our method's **parameter-sensitive analysis and visual effects**;

6. Provided the **specific task comparison** with our model.

---

Regarding the second-round rebuttal of the Reviewer Wi8X, it is worth noting that the **main contributions** of this study are the design of a **novel model** for texture removal tasks that provides **theoretical convergence analysis**. We have not focused on the innovations in the network modules. Additionally, we leverage **various evaluation metrics** (no-reference and reference IQA metrics) to assess the model's performance. We hope these responses fully address the reviewer's concerns, and **all these improvements are incorporated into our revised manuscript while maintaining clarity and technical depth**.

---

We are grateful to the reviewers for helping us improve our manuscript’s quality and completeness.

---

### Meta-Review · Area_Chair_HqQJ · 2026-01-02

**Summary:**

This paper proposes a residual pyramid atrous filtering network that utilizes the error low-rank representation to better handle multi-scale texture removal and structural preservation in the filtering process. The concerns of reviewers include the limited applications on real-world cases, strong theoretical assumption, the missing analysis of the computational efficiency and parameters w.r.t. the combination of dilation rate and low rank rank value, limited novelty (e.g.,  LSDC and DRL modules), and lack of explanation for experimental settings.

**Reviewer Concerns:**

The major concerns of reviewers include the limited applications on real-world cases, strong theoretical assumption, the missing analysis of the computational efficiency and parameters w.r.t. the combination of dilation rate and low rank rank value, limited novelty (e.g.,  LSDC and DRL modules), and lack of explanation for experimental settings.

**Reviewer Scores:**

The concerns of the limited applications on real-world cases, limited novelty (e.g.,  LSDC and DRL modules), and lack of explanation for experimental settings are not solved well.

---

### Decision · Program_Chairs · 2026-01-26

Reject